# Adversarial Moment-Matching Distillation of Large Language Models

**Chen Jia**
SI-TECH Information Technology
`jiachenwestlake@gmail.com`

## Abstract

Knowledge distillation (KD) has been shown to be highly effective in guiding a student model with a larger teacher model and achieving practical benefits in improving the computational and memory efficiency for large language models (LLMs). State-of-the-art KD methods for LLMs mostly rely on minimizing explicit distribution distance between teacher and student probability predictions. Instead of optimizing these mandatory behavior cloning objectives, we explore an imitation learning strategy for KD of LLMs. In particular, we minimize the imitation gap by matching the action-value moments of the teacher's behavior from both on- and off-policy perspectives. To achieve this action-value moment-matching goal, we propose an adversarial training algorithm to jointly estimate the moment-matching distance and optimize the student policy to minimize it. Results from both task-agnostic instruction-following experiments and task-specific experiments demonstrate the effectiveness of our method and achieve new state-of-the-art performance.

## 1 Introduction

Large language models (LLMs) like GPT-4 [1] and LLaMA [35] have revolutionized natural language processing, significantly enhancing the quality of text generation across various tasks. This success is largely due to the extensive scale of training data and the substantial increase in model parameters [19]. However, the high computational and memory requirements of these models present significant challenges for practical deployment. To address these issues, knowledge distillation (KD) [16] has emerged as a key technique. KD involves transferring knowledge from a large, complex teacher model to a smaller, more efficient student model, thereby maintaining high performance while reducing resource demands. Most distillation methods for auto-regressive text generation models, including LLMs, employ metrics of probability distribution distance, such as Kullback-Leibler (KL) divergence [20] and reverse KL divergence [14], aiming to align the token-level probability distributions between the teacher and student models.

The distribution matching-based distillation methods can be viewed as behavior cloning on a decision-making problem from the perspective of imitation learning [24, 14, 2]. Based on this concept, early works based on the teacher-generated outputs [20] or a supervised dataset [30] can be viewed as an *off-policy* approach. Recent works further incorporate an *on-policy* approach, training the student on its self-generated outputs [24], using KL-based divergence [14, 2, 21] and total variation (TV) distance [39]. Accordingly, such distribution matching-based methods face the sub-optimality problem. The objective functions aimed at aligning the probability distributions between the teacher and student models can be straightforward but cannot fully capture the goal of distilling language knowledge. First, intuitively, the correct output for an input can vary, and thus behavior cloning cannot capture the full knowledge of a teacher. Besides, there is no standardized definition for the quality of a generated output given an input, which makes it difficult to define the objective of knowledge distillation. This

38th Conference on Neural Information Processing Systems (NeurIPS 2024).

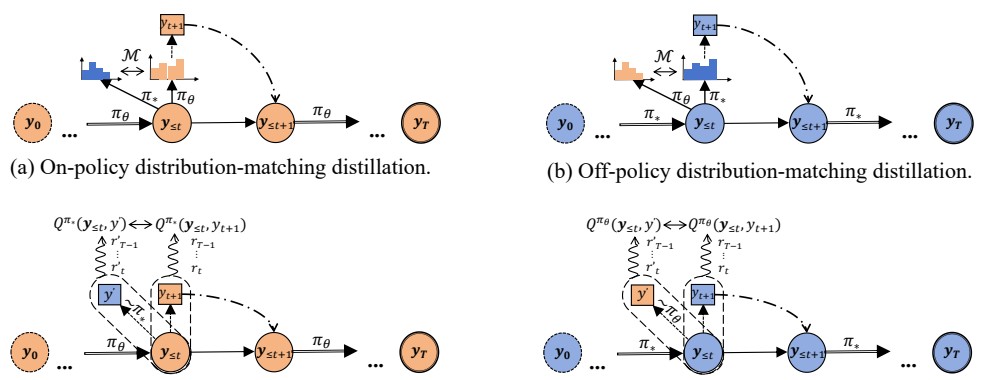

(a) On-policy distribution-matching distillation.

(b) Off-policy distribution-matching distillation.

(c) On-policy $Q$-value moment-match. distillation (**ours**).

(d) Off-policy $Q$-value moment-match. distillation (**ours**).

Figure 1: The comparison between the distribution-matching-based distillation and the action-value moment-matching distillation is outlined. $\pi_\theta$ and $\pi_*$ denote the student policy and the teacher policy, respectively. For both on-policy (using student-generated outputs) and off-policy (using teacher-generated outputs) perspectives, our approach optimizes moment-matching of action-value functions ($Q$-functions) instead of minimizing the distribution distance measured by $\mathcal{M}$ = KL, RKL, TV, etc.

imposes a significant limitation on the generalization performance of the student model through distillation.

To address the aforementioned issues, we employ a reinforcement learning (RL) formulation for the auto-regressive text generation problem and utilize the definition of imitation gap to describe the high-level goal of knowledge distillation. Additionally, we address the imitation gap for KD by matching moments of the action-value function, which reflects the quality of token-level predictions for the entire output. In addressing the action-value function, we adopt the approach of Swamy et al. [33], considering a two-player minimax game between the language policy and the action-value functions, aiming to minimize an upper bound of the moment-matching objective. For this purpose, we introduce an adversarial training algorithm based on the policy gradient to jointly optimize the on-/off-policy objectives. Figure 1 illustrates the overall approach.

Theoretically, we compare the moment-matching objective with other distribution-matching measurements such as step-wise TV distance and analyze the convergence rate of our algorithm to an $\epsilon$-accurate stationary point for optimization. Empirically, we evaluate our approach on both the instruction-following dataset and three task-specific datasets for text summarization, machine translation, and commonsense reasoning. Results demonstrate that the proposed adversarial moment-matching approach effectively optimizes the moment-matching distance of the imitation gap and outperforms state-of-the-art KD methods and a range of distribution-matching-based methods. The code and implementation are released at `https://github.com/jiachenwestlake/MMKD`.

## 2 Related Work

**Distillation of large language models.** There has been an increasing interest in knowledge distillation (KD) of auto-regressive LMs, especially concerning large language models (LLMs) [41, 42]. This process effectively transfers elicited knowledge from teacher LLMs to smaller student models, aiming to compress the large size of neural network parameters and make LLMs more efficient. Sequence-level KD (SeqKD) [20] is a variation of supervised fine-tuning (SFT) in KD. It can be viewed as the simplest method for distillation of black-box LLMs by fine-tuning the student model with teacher-generated outputs. This method has been extensively used for LLMs and has achieved success [34, 6]. In contrast, distillation of white-box LLMs can make full use of internal information of the teacher model, such as logits [30, 39] and hidden states [23], for distribution alignment, making it more effective and efficient for KD. However, unlike previous work that explicitly clones the distribution of teacher LLMs into student models, this work learns an auxiliary $Q$-value function to guide KD.

**Distillation via distribution matching.** Most promising results in the distillation of white-box LLMs are achieved by minimizing divergence between the probability distributions of the teacher model

and student models. Kullback-Leibler (KL) divergence, reverse Kullback-Leibler (RKL) divergence, and Jensen–Shannon (JS) divergence are three widely used KD objectives for auto-regressive LMs [39, 14, 2, 21, 41]. Wen et al. [39] have shown the equivalent formulations of sequence-level KL, RKL, JS divergences, and the step-wise terms. Additionally, they also present the strong performance of step-wise total variation (TV) distance for KD, which can upper bound the sequence-level term. As a result, most recent works focus on on-policy approaches for KD [2] and combine the real-time-generated outputs by students (on-policy) with the real-time-generated outputs by teachers (or from supervised datasets) (off-policy). Following this line, Gu et al. [14] further propose a policy gradient-based method to address the high variance issues of RKL-based methods while Ko et al. [21] propose a more efficient and effective method using a skew KL divergence loss and an adaptive off-policy approach. We also focus on a combination of on-policy and off-policy objectives for KD, but we introduce a more sophisticated moment-matching approach instead of directly using the well-studied distribution-matching metrics such as KL, RKL, JS divergences, and TV distance.

**Distillation via reinforcement learning.** In a common formulation of RL in text generation [44, 26, 15], an auto-regressive model can be viewed as a language policy, making decisions on the next token (`action`) based on the currently generated sequence (`state`). From this perspective, KD corresponds to behavior cloning in imitation learning [20, 7, 14, 2]. For imitation learning in text generation, early works such as SeqGAN [44] and TextGAIL [40] utilize a generative adversarial framework to balance between the reward model, optimized by discriminating generated/real-word text, and the language policy, optimized by policy gradient-based methods using the reward model. Existing work on KD via imitation learning refers to ImitKD [24], which optimizes the student policy by learning from demonstrations of the teacher model. RL-based distillation can also be especially relevant for leveraging the feedback from the teacher to train student models [4, 9], in which teacher models are used to generate the feedback data for training a reward model. We build our method upon an RL-based imitation learning framework. However, unlike previous work [20, 14, 2], we propose an adversarial moment-matching approach to enhance behavior cloning.

## 3 Method

### 3.1 Notations and Definitions

In this section, we consider the text generation task as a decision-making process and give a corresponding reinforcement learning (RL) formulation.

**Text generation.** Given an input $\boldsymbol{x}$, the auto-regressive generation task in our work aims to generate a sequence of tokens as the output $(y_1, \ldots, y_T)$, where $y_t$ comes from a vocabulary $\mathcal{V}$. For simplicity, we define $\boldsymbol{y} = (y_0, y_1, \ldots, y_T)$ as the full input-output sequence, where $y_0 = \boldsymbol{x}$ denotes the input. The generator is modeled by a conditional probability distribution $p_\theta(\boldsymbol{y}|\boldsymbol{x}) = \Pi_{t=0}^{T-1} p_\theta(y_{t+1}|\boldsymbol{y}_{\leq t})$, where $\boldsymbol{y}_{\leq t}$ denotes the prefix $(y_0, y_1, \ldots, y_t)$, $t \in \{0, 1, \ldots, T-1\}$.

**RL formulation.** We model text generation as a finite-horizon, time-independent Markov decision process. At each time step $t \in \{0, \ldots, T-1\}$, the policy $\pi_\theta$ takes an `action` $(t)$: $y_{t+1} \in \mathcal{V}$ based on the current `state` $(t)$: $\boldsymbol{y}_{\leq t} \in \mathcal{Y}$, transits to the next `state` $(t+1)$: $\boldsymbol{y}_{\leq t+1} \in \mathcal{Y}$ and receives a `reward` $(t)$: $r(\boldsymbol{y}_{\leq t}, y_{t+1})$ by a reward function $r : \mathcal{Y} \times \mathcal{V} \to \mathbb{R}$. The policy corresponds to the generation model $\pi_\theta(y_{t+1}|\boldsymbol{y}_{\leq t}) = p_\theta(y_{t+1}|\boldsymbol{y}_{\leq t})$. We focus on a (conditional) trajectory $\{y_1, \boldsymbol{y}_{\leq 1}, y_2, \ldots, \boldsymbol{y}_{\leq T-1}, y_T\} =: \tau \sim \pi_\theta|\boldsymbol{x}$ which refers to a sequence of state-action pairs generated by given an initial state $y_0 = \boldsymbol{x} \sim p_{\boldsymbol{x}}$ and then repeatedly sampling an action $y_{t+1} \sim \pi_\theta(\cdot|\boldsymbol{y}_{\leq t})$ and obtain the next state $\boldsymbol{y}_{\leq t+1} \sim T(\cdot|\boldsymbol{y}_{\leq t}, y_{t+1})$[1] for $T$ time steps. In such case, the probability of a (conditional) trajectory is formally represented as $p(\tau|\boldsymbol{x}, \pi_\theta) = \Pi_{t=0}^{T-1} T(\boldsymbol{y}_{\leq t+1}|\boldsymbol{y}_{\leq t}, y_{t+1})\pi_\theta(y_{t+1}|\boldsymbol{y}_{\leq t})$. We also define our value function and $Q$-value function as $V^{\pi_\theta}(\boldsymbol{y}_{\leq t}) = \mathbb{E}_{\tau_{(t)} \sim \pi_\theta|\boldsymbol{y}_{\leq t}} \left[ \sum_{t'=t}^{T-1} \gamma^{t'-t} r(\boldsymbol{y}_{\leq t'}, y_{t'+1}) \right]$ and $Q^{\pi_\theta}(\boldsymbol{y}_{\leq t}, y_{t+1}) = \mathbb{E}_{\tau_{(t)} \sim \pi_\theta|\boldsymbol{y}_{\leq t}, y_{t+1}} \left[ \sum_{t'=t}^{T-1} \gamma^{t'-t} r(\boldsymbol{y}_{\leq t'}, y_{t'+1}) \right]$, where $\gamma \in (0, 1)$ denotes the discounting factor. We define the RL objective in our generation task to maximize the performance $J(\pi_\theta) = \mathbb{E}_{\boldsymbol{x} \sim p_{\boldsymbol{x}}} \mathbb{E}_{\tau \sim \pi_\theta|\boldsymbol{x}} \left[ \sum_{t=0}^{T-1} \gamma^t r(\boldsymbol{y}_{\leq t}, y_{t+1}) \right]$.

---

[1]In text generation, the state-transition is commonly assumed to be deterministic [44, 26], i.e., $T(\boldsymbol{y}_{\leq t+1}|\boldsymbol{y}_{\leq t}, y_{t+1}) = 1$.

## 3.2 Knowledge Distillation as Moment-Matching Imitation Learning

Based on the RL formulation of auto-regressive generation, we can view the goal of knowledge distillation at a high-level as to bridge the performance gap between the teacher policy and the student policy.

**Definition 1 (Imitation gap).** *We define the imitation gap between the teacher policy and student policy as:*

$$J(\pi_*) - J(\pi_\theta) = \mathop{\mathbb{E}}_{\substack{\boldsymbol{x} \sim p_{\boldsymbol{x}} \\ \tau \sim \pi_* | \boldsymbol{x}}} \left[ \sum_{t=0}^{T-1} \gamma^t r(\boldsymbol{y}_{\leq t}, y_{t+1}) \right] - \mathop{\mathbb{E}}_{\substack{\boldsymbol{x} \sim p_{\boldsymbol{x}} \\ \tau \sim \pi_\theta | \boldsymbol{x}}} \left[ \sum_{t=0}^{T-1} \gamma^t r(\boldsymbol{y}_{\leq t}, y_{t+1}) \right], \quad (1)$$

From the perspective of imitation learning [33, 32], the objective of distillation from the teacher policy $\pi_*$ to the student policy $\pi_\theta$ can be represented as to minimize the imitation gap of Eq. (1) w.r.t. the parameters of student policy $\theta$. A direct idea from Eq. (1) is to use moment matching over the reward to optimize the imitation gap [33]. However, we actually care about the long-term reward, at each time step, we should consider the accumulated reward in the future output rather than the immediate reward to the fitness of previous tokens (prefix). To this end, we can alternatively use the $Q$-value function (def. in §3.1) for each timestep to represent the overall reward from the current timestep to the last timestep. Similar to [33], we can apply the Performance Difference Lemma (PDL) [18, 3, 33] to expand the imitation gap in Eq. (1) into either off-policy or on-policy expressions.

**Proposition 1 (Off-policy bound of imitation gap [33]).** *Let $\mathcal{F}_Q$ denote the set of $Q$-value functions induced by sampling actions from $\pi_\theta$, then we have:*

$$J(\pi_*) - J(\pi_\theta) \leq \sup_{f \in \mathcal{F}_Q} \underbrace{\mathop{\mathbb{E}}_{\substack{\boldsymbol{x} \sim p_{\boldsymbol{x}} \\ \tau \sim \pi_* | \boldsymbol{x}}} \left[ \sum_{t=0}^{T-1} \gamma^t \left( f(\boldsymbol{y}_{\leq t}, y_{t+1}) - \mathop{\mathbb{E}}_{y \sim \pi_\theta(\cdot | \boldsymbol{y}_{\leq t})} \left[ f(\boldsymbol{y}_{\leq t}, y) \right] \right) \right]}_{=:\mathcal{L}^{\mathrm{off}}(\pi_\theta, f)} \quad (2)$$

*In the following sections, we will use $\mathcal{L}^{\mathrm{off}}(\pi_\theta, f)$ to represent the off-policy moment-matching objective of mitation learning for KD.*

The off-policy moment-matching objective in Proposition 1 only requires a collected dataset of teacher-generated trajectories to be evaluated and minimized.

**Proposition 2 (On-policy bound of imitation gap [33]).** *Let $\mathcal{F}_{Q_*}$ denote the set of $Q$-value functions induced by sampling actions from $\pi_*$, then we have:*

$$J(\pi_*) - J(\pi_\theta) \leq \sup_{f \in \mathcal{F}_{Q_*}} \underbrace{\mathop{\mathbb{E}}_{\substack{\boldsymbol{x} \sim p_{\boldsymbol{x}} \\ \tau \sim \pi_\theta | \boldsymbol{x}}} \left[ \sum_{t=0}^{T-1} \gamma^t \left( \mathop{\mathbb{E}}_{y \sim \pi_*(\cdot | \boldsymbol{y}_{\leq t})} \left[ f(\boldsymbol{y}_{\leq t}, y) \right] - f(\boldsymbol{y}_{\leq t}, y_{t+1}) \right) \right]}_{=:\mathcal{L}^{\mathrm{on}}(\pi_\theta, f)} \quad (3)$$

*In the following sections, we will use $\mathcal{L}^{\mathrm{on}}(\pi_\theta, f)$ to represent the on-policy moment-matching objective of an imitation learning for KD.*

*Proof.* See Appendix A.1 and Appendix A.2 for the complete derivations of Proposition 1 and Proposition 2, respectively. ∎

It is notable from Proposition 2 that the on-policy moment-matching objective requires interactions with the teacher to tell us what action they would take in any state visited by the student as well as on-policy samples from the student's current policy $\tau \sim \pi_\theta | \boldsymbol{x}$.

In the remaining content of this section, we will explore the relationship between the moment-matching objectives and the existing distribution-matching objectives [39]. At the beginning, we draw a general formulation of the state-of-the-art methods for distillation of LLMs [39, 14, 2, 21] that rely on distribution-matching between the student's and teacher's predictions, through minimizing the step-wise probability distribution distance between the teacher policy and student policy.

**Definition 2** (**Generalized step-wise distribution distance**). *The off-policy and on-policy versions are defined as follows,*

$$d_{\mathcal{M}}^{\mathrm{off}}(\pi_\theta, \pi_*) := \underset{\substack{\boldsymbol{x} \sim p_{\boldsymbol{x}} \\ \tau \sim \pi_* | \boldsymbol{x}}}{\mathbb{E}} \left[ \sum_{t=0}^{T-1} \gamma^t \mathcal{M}(\pi_*(\cdot|\boldsymbol{y}_{\leq t}), \pi_\theta(\cdot|\boldsymbol{y}_{\leq t})) \right]; \tag{4}$$

$$d_{\mathcal{M}}^{\mathrm{on}}(\pi_\theta, \pi_*) := \underset{\substack{\boldsymbol{x} \sim p_{\boldsymbol{x}} \\ \tau \sim \pi_\theta | \boldsymbol{x}}}{\mathbb{E}} \left[ \sum_{t=0}^{T-1} \gamma^t \mathcal{M}(\pi_*(\cdot|\boldsymbol{y}_{\leq t}), \pi_\theta(\cdot|\boldsymbol{y}_{\leq t})) \right], \tag{5}$$

*where $\mathcal{M}(\cdot, \cdot)$ denotes a distribution distance, consisting of total variation (TV) distance [39] and Kullback-Leibler (KL)-based divergence [14, 2]. Detailed definitions for these distances refer to Appendix A.3. For simplicity, we directly replace $\mathcal{M}$ with TV, KL, RKL, etc in the following sections.*

It is notable from Wen et al. [39] that the sequence-level KL, RKL and JS divergences can be equivalently represented as the step-wise terms, and the sequence-level TV distance can be upper bounded by the step-wise terms, which can be actually implemented by algorithms. To make a connection with the step-wise distribution distance (Definition 2), we use the following definition.

**Definition 3** (**Distribution-matching formulation of moment-matching objectives**). *Based on Definition 2, we can re-formulate the off-policy and on-policy moment-matching (MM) objectives (Proposition 1 and Proposition 2, respectively) via step-wise distribution-matching, which can be defined as $d_{\mathrm{MM}}^{\mathrm{off}}(\pi_\theta, \pi_*)$ and $d_{\mathrm{MM}}^{\mathrm{on}}(\pi_\theta, \pi_*)$ respectively, where the distance metric $\mathrm{MM}(\cdot, \cdot)$ can be defined as follows,*

$$\mathrm{MM}^{\mathrm{off(on)}}(\pi_*(\cdot|\boldsymbol{y}_{\leq t}), \pi_\theta(\cdot|\boldsymbol{y}_{\leq t})) = \underset{y \sim \pi_*(\cdot|\boldsymbol{y}_{\leq t})}{\mathbb{E}} \left[ f_*^{\mathrm{off(on)}}(\boldsymbol{y}_{\leq t}, y) \right] - \underset{y \sim \pi_\theta(\cdot|\boldsymbol{y}_{\leq t})}{\mathbb{E}} \left[ f_*^{\mathrm{off(on)}}(\boldsymbol{y}_{\leq t}, y) \right],$$

$$\textbf{Off-policy: } f_*^{\mathrm{off}} = \underset{f \in \mathcal{F}_Q}{\arg\max} \, \mathcal{L}^{\mathrm{off}}(\pi_\theta, f); \quad \textbf{On-policy: } f_*^{\mathrm{on}} = \underset{f \in \mathcal{F}_{Q_*}}{\arg\max} \, \mathcal{L}^{\mathrm{on}}(\pi_\theta, f), \tag{6}$$

*where $\mathcal{L}^{\mathrm{off}}(\pi_\theta, f)$ and $\mathcal{L}^{\mathrm{on}}(\pi_\theta, f)$ denote the off-policy and on-policy moment-matching objectives, which are defined in Proposition 1 and Proposition 2, respectively.*

Under Definition 3, we observe that the main difference between the moment-matching objectives and other step-wise distribution distance, e.g., TV distance and KL-based divergences in formulation comes from the optimal $Q$-value function $f_*^{\mathrm{off(on)}}$, aiming to maximize the discrepancy of its expectations based on $\pi_*(\cdot|\boldsymbol{y}_{\leq t})$ *v.s.* $\pi_\theta(\cdot|\boldsymbol{y}_{\leq t})$ for each step $t \in \{0, 1, \ldots, T-1\}$. To look deeper, we draw a connection between the moment-matching objectives and step-wise TV distance using the following corollary.

**Theorem 1** (**Relationship between moment-matching objective and TV distance**). *Under a constrain of uniform boundness on the class of $Q$-value functions for off-/on-policy learning: $\mathcal{F}_Q = \mathcal{F}_{Q_*} = \{f : \|f\|_\infty \leq 1\}$, the moment-matching objectives in Proposition 1 and Proposition 2 can be upper-bounded by the step-wise TV distance, Formally, we have*

$$J(\pi_*) - J(\pi_\theta) \leq \sup_{f:\|f\|_\infty \leq 1} \mathcal{L}^{\mathrm{off}}(\pi_\theta, f) \leq 2d_{\mathrm{TV}}^{\mathrm{off}}(\pi_\theta, \pi_*); \tag{7}$$

$$J(\pi_*) - J(\pi_\theta) \leq \sup_{f:\|f\|_\infty \leq 1} \mathcal{L}^{\mathrm{on}}(\pi_\theta, f) \leq 2d_{\mathrm{TV}}^{\mathrm{on}}(\pi_\theta, \pi_*), \tag{8}$$

*for the off-policy and on-policy perspectives, respectively.*

*Proof.* See Appendix A.4 for the complete derivation. $\square$

We can observe from Theorem 1 that minimizing the step-wise TV distance can achieve sub-optimal results compared to optimizing the moment-matching objectives $\mathcal{L}^{\mathrm{off}}(\pi_\theta, f)$, $\mathcal{L}^{\mathrm{on}}(\pi_\theta, f)$ for off-policy and on-policy imitation learning, which are defined in Proposition 1 and Proposition 2, respectively. Thus, optimizing the moment-matching objectives can potentially achieve better optimization results for imitation learning.

**Algorithm 1:** Adversarial training procedure

---

**Input:** Dataset $\mathcal{D}_{\boldsymbol{xy}}$ with inputs and ground-truth outputs
Teacher policy $\pi_*$; Student policy $\pi_\theta$ with initial parameters $\theta$ pretrained on $\mathcal{D}_{\boldsymbol{xy}}$; Off-policy $Q$-value
function $f_{\phi_1}$ and on-policy $Q$-value function $f_{\phi_2}$ with initial parameters $\phi_1$ and $\phi_2$, respectively;
Step sizes $K$ (outer), $N$ (inner); Learning rate $\eta$; Controlling factor $\alpha$; Off-/on-policy combination factor $\beta$
**Output:** The optimized student policy $\pi_{\theta_*}$
**for** $k = 0, 1, 2, \ldots, K - 1$ **do**
    **for** $n = 0, 1, 2, \ldots, N - 1$ **do**
        Sample an input $\boldsymbol{x} \sim \mathcal{D}_{\boldsymbol{x}}$ and generate an trajectory $\tau^{\text{off}} \sim \pi_*|\boldsymbol{x}$
        $\phi_1 \leftarrow \phi_1 + \alpha\beta\eta\nabla_{\phi_1}\hat{\mathcal{L}}^{\text{off}}(\tau^{\text{off}}, \theta_k, f_{\phi_1})$             $\triangleright$ maximize $\mathcal{L}^{\text{off}}(\pi_{\theta_k}, f_{\phi_1})$ in Eq. (9)
        Sample an input $\boldsymbol{x} \sim \mathcal{D}_{\boldsymbol{x}}$ and generate an trajectory $\tau^{\text{on}} \sim \pi_\theta|\boldsymbol{x}$
        $\phi_2 \leftarrow \phi_2 + \alpha(1 - \beta)\eta\nabla_{\phi_2}\hat{\mathcal{L}}^{\text{on}}(\tau^{\text{on}}, \theta_k, f_{\phi_2})$        $\triangleright$ maximize $\mathcal{L}^{\text{on}}(\pi_{\theta_k}, f_{\phi_2})$ in Eq. (9)
    **end**
    Sample an input $\boldsymbol{x}_k \sim \mathcal{D}_{\boldsymbol{x}}$ and generate trajectories $\tau_k^{\text{off}} \sim \pi_*|\boldsymbol{x}_k$ and $\tau_k^{\text{on}} \sim \pi_\theta|\boldsymbol{x}_k$
    $\theta_{k+1} \leftarrow \theta_k - \eta\left(-\beta\hat{\mathcal{G}}^{\text{off}}(\tau_k^{\text{off}}, \theta_k) + (1 - \beta)\hat{\mathcal{G}}^{\text{on}}(\tau_k^{\text{on}}, \theta_k)\right)$    $\triangleright$ minimize $\mathcal{L}(\pi_\theta, f_{\phi_1}, f_{\phi_2})$ in Eq. (9)
**end**

---

### 3.3 Adversarial Training Algorithm

**Optimization objective.** As shown in previous work [14, 2, 21] incorporating both the off-policy and on-policy distillation benefits effectiveness and efficiency. We thus consider a training objective to jointly minimize the off-policy moment-matching objective in Proposition 1 and the on-policy moment-matching objective in Proposition 2. Both the off-/on-policy objectives can be optimized by viewing the learning procedure as solving a game. More specifically, we consider a two-player minimax game between the student policy and the $Q$-value functions. To this end, we initialize two small networks of a single-layer MLP to estimate the off-/on-policy $Q$-value functions, respectively. For example in a causal/seq-to-seq LM, the $Q$-value estimate module can be represented as $f_{\phi_{1(2)}}(\boldsymbol{y}_{\leq t}, y) = (\mathbf{h}_t^{\pi_\theta} + \mathbf{v}_y^{\text{off(on)}})^\top \mathbf{w}_y^{\text{off(on)}}$ for any action token $y \in \mathcal{V}$. This estimates the $Q$-value function by taking the current $t \in \{0, 1, \ldots, T - 1\}$ hidden step of a policy network $\mathbf{h}_t^{\pi_\theta} \in \mathbb{R}^H$ (for next token prediction) to combine with the feature vector of the token $\mathbf{v}_y^{\text{off(on)}} \in \mathbb{R}^H$ with a linear transformation by $\mathbf{w}_y^{\text{off(on)}} \in \mathbb{R}^H$ for off(on)-policy learning. Here, $H$ represents the hidden size and the additional parameter cost is $\mathcal{O}(H|\mathcal{V}|)$ for $Q$-value estimation. Finally, combining off- and on-policy objectives with a factor $\beta \in (0, 1)$, the optimization problem can be represented as follows,

$$\min_{\theta \in \Theta} \max_{\phi_1, \phi_2 \in \Phi} \underbrace{\beta\mathcal{L}^{\text{off}}(\pi_\theta, f_{\phi_1}) + (1 - \beta)\mathcal{L}^{\text{on}}(\pi_\theta, f_{\phi_2})}_{=:\mathcal{L}(\pi_\theta, f_{\phi_1}, f_{\phi_2})}, \tag{9}$$

where $\mathcal{L}(\pi_\theta, f_{\phi_1}, f_{\phi_2})$ represents the overall training objective. To minimize the objective w.r.t the policy parameters $\theta$, we use a policy gradient approach and derive the policy gradient in Appendix A.5, formally represented as follows,

$$\nabla\mathcal{L}(\pi_\theta, f_{\phi_1}, f_{\phi_2}) = \mathop{\mathbb{E}}_{\boldsymbol{x} \sim p_{\boldsymbol{x}}}\left[-\beta\mathop{\mathbb{E}}_{\tau \sim \pi_*|\boldsymbol{x}}\left[\hat{\mathcal{G}}^{\text{off}}(\tau, \theta)\right] + (1 - \beta)\mathop{\mathbb{E}}_{\tau' \sim \pi_\theta|\boldsymbol{x}}\left[\hat{\mathcal{G}}^{\text{on}}(\tau', \theta)\right]\right]$$

$$\text{s.t.} \quad \hat{\mathcal{G}}^{\text{off}}(\tau, \theta) = \sum_{t=0}^{T-1}\gamma^t\mathop{\mathbb{E}}_{y \sim \pi_\theta(\cdot|\boldsymbol{y}_{\leq t})}\left[\nabla\log\pi_\theta(y|\boldsymbol{y}_{\leq t})f_{\phi_1}(\boldsymbol{y}_{\leq t}, y)\right]; \tag{10}$$

$$\hat{\mathcal{G}}^{\text{on}}(\tau', \theta) = \sum_{t=0}^{T-1}\gamma^t\nabla\log\pi_\theta(y'_{t+1}|\boldsymbol{y}'_{\leq t})\hat{Q}_{f_{\phi_2}}(\boldsymbol{y}'_{\leq t}, y'_{t+1}),$$

where $\hat{Q}_{f_{\phi_2}} : \mathcal{Y} \times \mathcal{V} \to \mathbb{R}$ denotes the empirical $Q$-value defined in Eq. (21). Besides, we use stochastic gradient ascent (SGA) to maximize the objective of $\mathcal{L}(\pi_\theta, f_{\phi_1}, f_{\phi_2})$ w.r.t. parameters of the on-policy $Q$-value function $\phi_1$ and parameters of the off-policy $Q$-value function $\phi_2$.

**Training procedure.** The goal is to achieve an equilibrium between minimizing the objective w.r.t. the parameters of student policy $\theta \in \Theta$ and maximizing the objective w.r.t. the parameters of on-policy and off-policy $Q$-value functions $\phi_1, \phi_2 \in \Phi$, formally defined as $\min_\theta \max_{\phi_1, \phi_2} \mathcal{L}(\pi_\theta, f_{\phi_1}, f_{\phi_2})$

(Eq. (9)). To this end, we use an adversarial training strategy in Algorithm 1, by starting from a student model fine-tuned on a dataset $\mathcal{D}_{\boldsymbol{xy}}$. In the training algorithm, we iteratively maximize the objective w.r.t. the parameters of $Q$-value functions $f_{\phi_1}, f_{\phi_2}$ and simultaneously minimize the objective w.r.t. the parameters of student policy $\pi_\theta$. In each iteration of policy updating, we first perform $N$ steps of stochastic gradient ascent (SGA) w.r.t. the parameters of $Q$-value functions $\phi_1, \phi_2$. Then, the parameters of student policy $\theta$ are updated by stochastic gradient descent (SGD) with the estimated policy gradient with sampling policy gradients.

### 3.4 Convergence Analysis

We further provide a convergence analysis for the algorithm proposed in §3.3. To deal with the challenges of non-convexity by certain reward structures, the algorithm is expected to obtain an $\epsilon$-accurate stationary point of the policy parameters $\theta_* \in \Theta$, satisfying that $\mathbb{E}[\|\overline{\nabla}\mathcal{L}(\theta_*)\|^2] \leq \epsilon$. We focus on policy optimization and directly use the optimized off-/on-policy $Q$-value functions in each outer-loop iteration $k \in \{0, 1, \ldots, K-1\}$. We denote $\phi_1(\theta_k) = \arg\max_{\phi_1} \mathcal{L}^{\text{off}}(\pi_{\theta_k}, f_{\phi_1})$, $\phi_2(\theta_k) = \arg\max_{\phi_2} \mathcal{L}^{\text{on}}(\pi_{\theta_k}, f_{\phi_2})$ as the inner-loop optimized functions and use $\mathcal{L}(\theta_k) := \mathcal{L}(\pi_{\theta_k}, f_{\phi_1(\theta_k)}, f_{\phi_2(\theta_k)})$ (def. in Eq. (9)) for simplicity in this section. We start with the following standard assumption [45].

**Assumption 1.** *Suppose that the optimized $Q$-value functions and the parameterized policy $\pi_\theta$ satisfy the following conditions:*

*(i) The uniformly boundness of off/on-policy $Q$-value functions optimized by Algorithm 1, i.e., $\|f_{\phi_1}\|_\infty, \|f_{\phi_2}\|_\infty \leq 1$.*

*(ii) The B-Lipschitzness and the L-smoothness of the parameterized policy, i.e., for any state-action pair $(\boldsymbol{y}_{\leq t}, y_{t+1}) \in \mathcal{Y} \times \mathcal{V}$ at any time step $t \in \{0, 1, \ldots, T-1\}$,*

$$\|\nabla \log \pi_\theta(y_{t+1}|\boldsymbol{y}_{\leq t})\| \leq B, \text{ for any } \theta \in \Theta, \tag{11}$$

$$\|\nabla \log \pi_{\theta_1}(y_{t+1}|\boldsymbol{y}_{\leq t}) - \nabla \log \pi_{\theta_2}(y_{t+1}|\boldsymbol{y}_{\leq t})\| \leq L\|\theta_1 - \theta_2\|, \text{ for any } \theta_1, \theta_2 \in \Theta \tag{12}$$

**Theorem 2** (**Convergence rate of Algorithm 1 to stationary points**). *Let $\{\theta_k\}_{1 \leq k \leq K}$ be the sequence of parameters of the policy $\pi_{\theta_k}$ given by Algorithm 1. Let the learning rate $\eta = \frac{2}{B_{\mathcal{L}}}\sqrt{\frac{1-\gamma^T}{(1-\gamma)KL_{\mathcal{L}}}}$. Under Assumption 1, we have*

$$\min_{0 \leq k \leq K-1} \mathbb{E}\left[\|\nabla\mathcal{L}(\theta_k)\|^2\right] \leq \mathcal{O}\left(\frac{1}{\sqrt{K}}\right) \tag{13}$$

*Proof.* See Appendix A.6 for the complete derivation. $\qquad\square$

Theorem 2 illustrates that the output gradient norm square by Algorithm 1 can converge to a neighborhood around zero with the rate of $1/\sqrt{K}$. Furthermore, leveraging a sufficient number of training iterations $\mathcal{O}(\epsilon^{-2})$, Algorithm 1 can obtain an $\epsilon$-accurate stationary point. This leads to the following corollary on the computational complexity of the training procedure.

**Corollary 1** (**Computational complexity of Algorithm 1**). *We formalize the policy as a softmax function $\pi_\theta$ with a linear transformation:* $\text{softmax}(\theta\boldsymbol{y}_{\leq t})$ *for any $\boldsymbol{y}_{\leq t} \in \mathbb{R}^H$, where $\theta \in \mathbb{R}^{|\mathcal{V}| \times H}$ and $H$ denotes the hidden size. Then, to obtain an $\epsilon$-accurate stationary point by Algorithm 1, the complexity of gradient computation is $\mathcal{O}(\epsilon^{-2}T|\mathcal{V}|H(N + T + |\mathcal{V}|))$.*

*Proof.* See Appendix A.7 for the complete derivation. $\qquad\square$

Corollary 1 shows that Algorithm 1 has a polynomial computational complexity w.r.t $\epsilon^{-2}$, $N$, $|\mathcal{V}|$, $H$ and $T$, to obtain an $\epsilon$-accurate stationary point for optimizing the training objective in Eq. (9).

## 4 Experiments

We consider task-agnostic instruction-following experiments and task-specific experiments, including text summarization, machine translation, and commonsense reasoning. We compare our approach

Table 1: Comparison with state-of-the-art KD methods on the instruction-following dataset using fine-tuned OpenLLaMA-7B as the teacher and fine-tuned OpenLLaMA-3B as the student. We format **the best**, the second best and worse than SFT results. The results based on GPT-2 are available in Appendix C.1.

| Method | DollyEval | | SelfInst | | VicunaEval | | S-NI | UnNI |
|---|---|---|---|---|---|---|---|---|
| | GPT-4 | R-L | GPT-4 | R-L | GPT-4 | R-L | R-L | R-L |
| *OpenLLaMA2-7B (teacher)* | $58.8_{\pm1.2}$ | $32.5_{\pm0.4}$ | $56.7_{\pm0.8}$ | $21.6_{\pm0.2}$ | $46.2_{\pm0.6}$ | $22.6_{\pm0.5}$ | $36.3_{\pm0.5}$ | $38.5_{\pm0.2}$ |
| SFT (*student*) | $46.8_{\pm0.7}$ | $26.7_{\pm0.6}$ | $40.8_{\pm1.1}$ | $16.3_{\pm0.7}$ | $34.8_{\pm0.8}$ | $17.3_{\pm0.2}$ | $30.4_{\pm0.4}$ | $28.6_{\pm0.3}$ |
| KD [16] | $43.9_{\pm0.8}$ | $22.4_{\pm0.4}$ | $43.5_{\pm0.5}$ | $17.4_{\pm0.5}$ | $33.7_{\pm0.3}$ | $16.4_{\pm0.2}$ | $29.3_{\pm0.6}$ | $23.4_{\pm0.3}$ |
| SeqKD [20] | $50.2_{\pm0.6}$ | $26.2_{\pm0.4}$ | $46.8_{\pm0.3}$ | $15.8_{\pm0.5}$ | $38.8_{\pm1.2}$ | $18.0_{\pm0.6}$ | $29.7_{\pm0.3}$ | $27.8_{\pm0.1}$ |
| ImitKD [24] | $53.7_{\pm1.6}$ | $25.3_{\pm0.3}$ | $45.0_{\pm0.7}$ | $18.4_{\pm0.4}$ | $41.7_{\pm1.2}$ | $19.1_{\pm0.2}$ | $33.1_{\pm0.7}$ | $28.7_{\pm0.5}$ |
| MiniLLM [14] | $58.7_{\pm1.2}$ | $28.4_{\pm0.3}$ | $51.8_{\pm1.5}$ | $20.2_{\pm0.6}$ | $44.2_{\pm1.1}$ | $20.7_{\pm0.5}$ | $37.4_{\pm0.4}$ | $37.5_{\pm0.2}$ |
| GKD [2] | $57.6_{\pm1.0}$ | $27.5_{\pm0.3}$ | $52.4_{\pm1.2}$ | $20.9_{\pm0.3}$ | $45.5_{\pm0.8}$ | $19.3_{\pm0.5}$ | $36.8_{\pm0.6}$ | $34.8_{\pm0.3}$ |
| DistiLLM [21] | $59.2_{\pm1.2}$ | $29.5_{\pm0.2}$ | $53.4_{\pm1.0}$ | $20.8_{\pm0.7}$ | $46.3_{\pm0.9}$ | $20.4_{\pm0.3}$ | $37.2_{\pm0.1}$ | $38.2_{\pm0.1}$ |
| **Ours** | $59.8_{\pm0.8}$ | $30.7_{\pm0.4}$ | $54.2_{\pm1.2}$ | $21.7_{\pm0.5}$ | $47.8_{\pm0.7}$ | $21.4_{\pm0.4}$ | $38.7_{\pm0.4}$ | $39.1_{\pm0.3}$ |

with various KD baselines, including: SFT, which fine-tunes the student model on the supervised dataset $\mathcal{D}_{xy}$; KD [16], which uses KL divergence on the supervised dataset $\mathcal{D}_{xy}$; SeqKD [20], which applies SFT to the student model with teacher-generated outputs; ImitKD [24], which uses KL divergence on the student-generated outputs; MiniLLM [14], which uses RKL divergence with a policy gradient method; GKD [2], which uses JS divergence with an on-policy method; and DistiLLM [21], which uses an adaptive training method for off-policy optimization of a skew KL divergence. Additionally, we focus on step-wise distance optimization for KD and compare it with a range of well-known methods, including KL divergence, RKL divergence, JS divergence, and TV distance, as discussed by Wen et al. [39]. All the reported results are the average across three random seeds.

## 4.1 Task-Agnostic Distillation

**Experimental Setup.** We follow the previous works [14, 21] for the implementation of the instruction-following experiment, aiming to evaluate the distilled model's ability to handle diverse tasks presented in the form of instructions. We construct the training data from `databricks-dolly-15k` [8], where we randomly select 15K samples for training and equally split 500 samples for validation and testing. We evaluate the trained model on five instruction-following datasets: DollyEval, SelfInst [36], VicunaEval [6], S-NI [37], and UnNI [17]. Following the previous works [14, 21], we also add the OpenWebText [13] corpus, consisting of long-document plain text, for joint training with a language modeling task. This has been shown to effectively improve the performance of instruction tuning [14]. The evaluation metrics include ROUGE-L [25] and GPT-4 feedback with the same prompts as in [21]. More details on experimental setup refer to Appendix B.

**Main results.** Table 1 illustrates the instruction-following performances. Compared with the SFT baseline, which indicates the student model without KD, KD and SeqKD hardly improve the performances. This indicates that using only supervised datasets or teacher-generated outputs does not benefit the KD of large language models. In contrast, utilizing the student-generated outputs with KL divergence [2], RKL divergence [14], and JS divergence [2] shows effectiveness for KD in the instruction-following task. State-of-the-art methods [14, 2, 21] tend to combine the student-generated outputs with the teacher-generated output or supervised dataset to further improve the results of KD. This shows that a mixture optimization of both on-policy and off-policy objectives can effectively improve the KD performance of large language models on the instruction-following task. In particular, we use an adversarial moment-matching method and optimize both on-policy and off-policy objectives for KD, thus achieving the best results on five test datasets with both GPT-4 feedback and ROUGE-L evaluations.

## 4.2 Task-Specific Distillation

**Experimental Setup.** We evaluated the KD models on three tasks consisting of text summarization, machine translation, and reasoning. For the text summarization task, we follow Ko et al. [21] to conduct experiments on the SAMSum [12] dataset. For the machine translation tasks, we follow Ko et al. [21] to conduct experiments on the IWSLT'17 (en-de) [5] dataset. For the commonsense reasoning task, we conduct experiments on the StrategyQA dataset [11] with chain-of-thought augmentations

Table 2: Comparison with the state-of-the-art KD methods on text summarization, machine translation and commonsense reasoning datasets. We report the ROUGE-L, BLEU and accuracy for SAMSum, IWSLT'17 (en-de) and StrategyQA, respectively. We format **the best**, the second best and worse than SFT results.

| Method | SAMSum | | | IWSLT'17 (en-de) | | | StrategyQA | | |
|---|---|---|---|---|---|---|---|---|---|
| | T5-Small | T5-Base | T5-Large | T5-Small | T5-Base | T5-Large | T5-Small | T5-Base | T5-Large |
| *T5-XL (teacher)* | | $52.5_{\pm0.4}$ | | | $35.2_{\pm0.2}$ | | | $64.5_{\pm0.8}$ | |
| SFT (*student*) | $40.6_{\pm0.2}$ | $47.3_{\pm0.3}$ | $49.8_{\pm0.2}$ | $21.5_{\pm0.1}$ | $30.1_{\pm0.0}$ | $33.7_{\pm0.1}$ | $52.4_{\pm0.5}$ | $57.5_{\pm0.8}$ | $60.7_{\pm0.8}$ |
| KD [16] | $39.2_{\pm0.4}$ | $46.5_{\pm0.3}$ | $47.4_{\pm0.3}$ | $21.7_{\pm0.1}$ | $29.8_{\pm0.2}$ | $31.7_{\pm0.1}$ | $49.7_{\pm0.3}$ | $55.3_{\pm0.1}$ | $59.2_{\pm0.5}$ |
| SeqKD [20] | $39.7_{\pm0.3}$ | $47.7_{\pm0.5}$ | $49.3_{\pm0.4}$ | $21.2_{\pm0.3}$ | $29.2_{\pm0.2}$ | $32.9_{\pm0.5}$ | $50.6_{\pm0.7}$ | $57.5_{\pm1.1}$ | $61.5_{\pm0.8}$ |
| ImitKD [24] | $41.8_{\pm0.3}$ | $48.6_{\pm0.7}$ | $51.2_{\pm0.5}$ | $22.2_{\pm0.3}$ | $28.7_{\pm0.6}$ | $34.1_{\pm0.2}$ | $53.8_{\pm0.8}$ | $59.7_{\pm0.5}$ | $61.7_{\pm0.6}$ |
| GKD [2] | $42.1_{\pm0.3}$ | $48.2_{\pm0.5}$ | $51.7_{\pm0.4}$ | $22.7_{\pm0.2}$ | $31.2_{\pm0.1}$ | $34.7_{\pm0.2}$ | $55.6_{\pm0.4}$ | $60.3_{\pm0.5}$ | $63.6_{\pm0.3}$ |
| DistiLLM [21] | $42.6_{\pm0.2}$ | $49.4_{\pm0.6}$ | $52.1_{\pm0.4}$ | $22.5_{\pm0.1}$ | $30.8_{\pm0.2}$ | $35.5_{\pm0.1}$ | $56.3_{\pm0.3}$ | $61.2_{\pm0.7}$ | $62.8_{\pm0.2}$ |
| **Ours** | **$43.7_{\pm0.4}$** | **$50.4_{\pm0.3}$** | **$52.7_{\pm0.3}$** | **$23.7_{\pm0.1}$** | **$32.4_{\pm0.3}$** | **$36.0_{\pm0.2}$** | **$58.2_{\pm0.4}$** | **$62.9_{\pm0.3}$** | **$65.3_{\pm0.7}$** |

[38]. For all of the task-specific experiments, we use T5-XL [29] as the teacher model and T5-Large/-Base/-Small as the student model. For the machine translation experiments, we employ a multilingual pretrained model, mT5 [43], to build the methods. For evaluation, we use ROUGE-L [25], BLEU [27], and accuracy as the performance metrics on SAMSum, IWSLT'17 (en-de), and StrategyQA, respectively. More details about the experimental setup refer to Appendix B.

**Main results.** Table 2 displays the performances on three task-specific datasets. Since the original work of MiniLLM [14] does not consider these tasks, we thus do not make comparisons with MiniLLM. The performance trend is similar to the instruct-following results, revealing that KD of large language models for specific tasks also benefits from the combination of on-policy objectives with student-generated outputs and off-policy objectives with teacher-generated outputs or supervised datasets. Additionally, we observe that student models of different sizes all benefit from the KD methods to improve performance. Overall, our approach achieves the best results on all three task-specific datasets for student models of different sizes. This demonstrates the effectiveness of an adversarial moment-matching approach for KD of large language models on specific tasks.

### 4.3 Analysis on Step-Wise Distance Optimization

**Comparison with distribution matching.** We make comparisons with different step-wise distribution distances with a uniform formulation of Definition 2, considering the on-policy, off-policy objectives as well as the joint form. Results on four tasks with a default combination factor $\beta = 0.5$ are shown in Figure 2. More instruct-following results are available in Appendix C.2 and results with different values of off-/on-policy combination factor are available in Appendix C.5. Compared with the KL divergence, RKL divergence, JS divergence and total variation

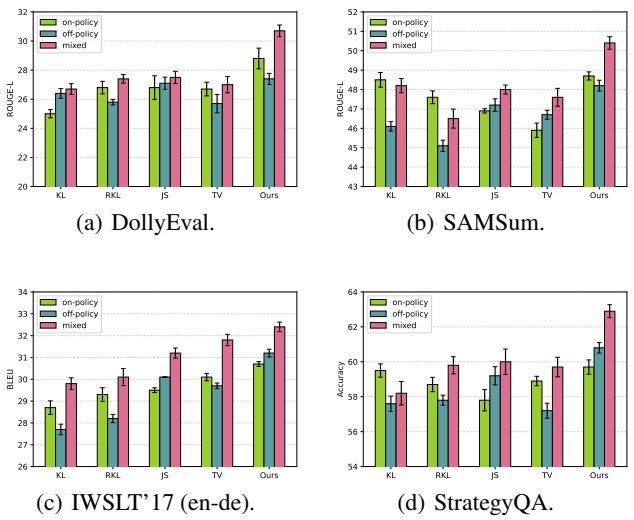

(a) DollyEval.      (b) SAMSum.

(c) IWSLT'17 (en-de).      (d) StrategyQA.

Figure 2: Performance of difference step-wise distribution distances.

distance, the proposed moment-matching distance achieves the best results under both the on-policy and off-policy training objectives, which shows that the proposed moment-matching approach is effective for KD of large language models. Besides, we observe that using a joint objective of both on-policy and off-policy can further significantly improve the performances. This shows that both on-policy and off-policy moment-matching objectives contribute to the minimization of the imitation gap and can thus benefit the KD of large language models.

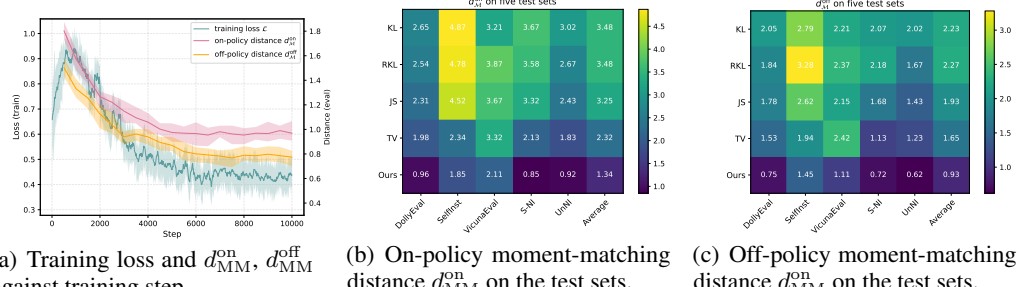

(a) Training loss and $d_{\text{MM}}^{\text{on}}$, $d_{\text{MM}}^{\text{off}}$ against training step.

(b) On-policy moment-matching distance $d_{\text{MM}}^{\text{on}}$ on the test sets.

(c) Off-policy moment-matching distance $d_{\text{MM}}^{\text{on}}$ on the test sets.

Figure 3: Adversarial training procedure for optimizing the on-policy and off-policy moment-matching distances $d_{\text{MM}}^{\text{on}}$, $d_{\text{MM}}^{\text{off}}$ on the instruction-following dataset.

**Adversarial training procedure.** We present the training loss and moment-matching distance against the adversarial training steps. As depicted in Figure 3 (a), the training loss initially increases within the first 0-1,000 steps, indicating that initially, the $Q$-value functions are stronger than the policy in maximizing the loss function $\mathcal{L}(\pi_\theta, f_{\phi_1}, f_{\phi_2})$ in Eq. (9). Concurrently, the policy gradient method contributes to minimizing the training loss, which eventually converges to a much lower stable value. Additionally, both the on-policy and off-policy moment-matching distances $d_{\text{MM}}^{\text{on}}$ and $d_{\text{MM}}^{\text{off}}$ decrease and eventually reach a low value with only minor fluctuations. For more results and details on experimental setups, please refer to Appendix C.3.

**Moment-matching distance optimization.** We further illustrate the on-policy moment-matching distance $d_{\text{MM}}^{\text{on}}$ and the off-policy moment-matching distance $d_{\text{MM}}^{\text{off}}$ (defined in Definition 3) optimized by different step-wise distances in Figure 3 (b) and (c), respectively. Interestingly, we observe that the total variation (TV) distance obtains the second-best results on average for both on-policy and off-policy distances. This finding suggests a similarity between the formulations of TV distance and moment-matching distances to some extent, as supported by the theoretical result of Theorem 1. Across all instruction-following test sets, our approach effectively optimizes both on-policy and off-policy moment-matching distances more than other step-wise distribution distances used in KD, including KL divergence, RKL divergence, JS divergence, and TV distance. This observation also underscores the effectiveness of our policy gradient methods. Extensive results on the task-specific datasets are available in Appendix C.4.

## 5    Conclusion

In this work, we investigated a moment-matching approach for knowledge distillation of large language models. Specifically, we formulated knowledge distillation from a perspective of imitation learning and derived both on-policy and off-policy bounds for the imitation gap between the teacher model and student model via moment-matching distance. Additionally, we proposed an adversarial training algorithm to simultaneously estimate and minimize the joint objective of on-policy and off-policy moment-matching distances. In experiments, we evaluated the proposed algorithm on four instruction-following datasets and three task-specific datasets, comparing it with a range of state-of-the-art KD methods as well as four well-studied step-wise distribution distances for KD of auto-regressive models. Results demonstrate that our approach can effectively leverage the policy gradient method to optimize the moment-matching distance and achieve the best results across all datasets.

**Limitations and future work.** The proposed adversarial training algorithm requires additional computational steps for the inner-loop gradient ascent, which may result in increased time complexity. Moreover, the proposed approach necessitates auxiliary networks to build the $Q$-value functions, which may incur additional memory costs. Besides, the experiments are conducted with limited LLM architectures, such as OpenLLaMA and T5. Therefore, in future work, we aim to enhance the time and memory efficiency of our approach, and evaluate the proposed approach on a wider range of architectures.

## Acknowledgements

We thank the anonymous reviewers for their helpful comments and suggestions. This work was supported by SI-TECH Information Technology Co., Ltd.

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

# A Proofs

## A.1 Proof of Proposition 1

*Proof.* Similar to the proof of Performance Difference Lemma (PDL) [18, 3, 33], we have

$$J(\pi_*) - J(\pi_\theta)$$

$$= \mathbb{E}_{\substack{\boldsymbol{x} \sim p_{\boldsymbol{x}} \\ \tau \sim \pi_* | \boldsymbol{x}}} \left[ \sum_{t=0}^{T-1} \gamma^t r(\boldsymbol{y}_{\leq t}, y_{t+1}) \right] - \mathbb{E}_{\boldsymbol{x} \sim p_{\boldsymbol{x}}} [V^{\pi_\theta}(\boldsymbol{x})]$$

$$= \mathbb{E}_{\substack{\boldsymbol{x} \sim p_{\boldsymbol{x}} \\ \tau \sim \pi_* | \boldsymbol{x}}} \left[ \sum_{t=0}^{T-1} \gamma^t \left( r(\boldsymbol{y}_{\leq t}, y_{t+1}) + V^{\pi_\theta}(\boldsymbol{y}_{\leq t}) - V^{\pi_\theta}(\boldsymbol{y}_{\leq t}) \right) \right] - \mathbb{E}_{\boldsymbol{x} \sim p_{\boldsymbol{x}}} [V^{\pi_\theta}(\boldsymbol{x})]$$

$$= \mathbb{E}_{\substack{\boldsymbol{x} \sim p_{\boldsymbol{x}} \\ \tau \sim \pi_* | \boldsymbol{x}}} \left[ \sum_{t=0}^{T-1} \gamma^t \left( r(\boldsymbol{y}_{\leq t}, y_{t+1}) + \gamma V^{\pi_\theta}(\boldsymbol{y}_{\leq t+1}) - V^{\pi_\theta}(\boldsymbol{y}_{\leq t}) \right) \right]$$

$$= \mathbb{E}_{\substack{\boldsymbol{x} \sim p_{\boldsymbol{x}} \\ \tau \sim \pi_* | \boldsymbol{x}}} \left[ \sum_{t=0}^{T-1} \gamma^t \left( r(\boldsymbol{y}_{\leq t}, y_{t+1}) + \gamma \mathbb{E}_{\boldsymbol{y}_{\leq t+1} \sim T(\cdot | \boldsymbol{y}_{\leq t}, y_{t+1})} \left[ V^{\pi_\theta}(\boldsymbol{y}_{\leq t+1}) \right] - V^{\pi_\theta}(\boldsymbol{y}_{\leq t}) \right) \right]$$

$$\overset{(i)}{=} \mathbb{E}_{\substack{\boldsymbol{x} \sim p_{\boldsymbol{x}} \\ \tau \sim \pi_* | \boldsymbol{x}}} \left[ \sum_{t=0}^{T-1} \gamma^t \left( Q^{\pi_\theta}(\boldsymbol{y}_{\leq t}, y_{t+1}) - V^{\pi_\theta}(\boldsymbol{y}_{\leq t}) \right) \right]$$

$$= \mathbb{E}_{\substack{\boldsymbol{x} \sim p_{\boldsymbol{x}} \\ \tau \sim \pi_* | \boldsymbol{x}}} \left[ \sum_{t=0}^{T-1} \gamma^t \left( Q^{\pi_\theta}(\boldsymbol{y}_{\leq t}, y_{t+1}) - \mathbb{E}_{y \sim \pi_\theta(\cdot | \boldsymbol{y}_{\leq t})} \left[ Q^{\pi_\theta}(\boldsymbol{y}_{\leq t}, y) \right] \right) \right]$$

$$\leq \sup_{f \in \mathcal{F}_Q} \mathbb{E}_{\substack{\boldsymbol{x} \sim p_{\boldsymbol{x}} \\ \tau \sim \pi_* | \boldsymbol{x}}} \left[ \sum_{t=0}^{T-1} \gamma^t \left( f(\boldsymbol{y}_{\leq t}, y_{t+1}) - \mathbb{E}_{y \sim \pi_\theta(\cdot | \boldsymbol{y}_{\leq t})} \left[ f(\boldsymbol{y}_{\leq t}, y) \right] \right) \right],$$

where $(i)$ follows from Bellman equation and noting that the transition probability $T(\cdot | \boldsymbol{y}_{\leq t}, y_{t+1})$ is deterministic in an auto-regressive text generation problem. This completes the proof. $\square$

## A.2 Proof of Proposition 2

*Proof.* Similar to the proof of Proposition 1, we have

$$J(\pi_*) - J(\pi_\theta)$$

$$= - \mathbb{E}_{\substack{\boldsymbol{x} \sim p_{\boldsymbol{x}} \\ \tau \sim \pi_\theta | \boldsymbol{x}}} \left[ \sum_{t=0}^{T-1} \gamma^t r(\boldsymbol{y}_{\leq t}, y_{t+1}) \right] + \mathbb{E}_{\boldsymbol{x} \sim p_{\boldsymbol{x}}} [V^{\pi_*}(\boldsymbol{x})]$$

$$= \mathbb{E}_{\substack{\boldsymbol{x} \sim p_{\boldsymbol{x}} \\ \tau \sim \pi_\theta | \boldsymbol{x}}} \left[ \sum_{t=0}^{T-1} \gamma^t \left( V^{\pi_*}(\boldsymbol{y}_{\leq t}) - \left( r(\boldsymbol{y}_{\leq t}, y_{t+1}) + V^{\pi_*}(\boldsymbol{y}_{\leq t}) \right) \right) \right] + \mathbb{E}_{\boldsymbol{x} \sim p_{\boldsymbol{x}}} [V^{\pi_*}(\boldsymbol{x})]$$

$$= \mathbb{E}_{\substack{\boldsymbol{x} \sim p_{\boldsymbol{x}} \\ \tau \sim \pi_\theta | \boldsymbol{x}}} \left[ \sum_{t=0}^{T-1} \gamma^t \left( V^{\pi_*}(\boldsymbol{y}_{\leq t}) - \left( r(\boldsymbol{y}_{\leq t}, y_{t+1}) + \gamma V^{\pi_*}(\boldsymbol{y}_{\leq t+1}) \right) \right) \right]$$

$$= \mathbb{E}_{\substack{\boldsymbol{x} \sim p_{\boldsymbol{x}} \\ \tau \sim \pi_\theta | \boldsymbol{x}}} \left[ \sum_{t=0}^{T-1} \gamma^t \left( V^{\pi_*}(\boldsymbol{y}_{\leq t}) - \left( r(\boldsymbol{y}_{\leq t}, y_{t+1}) + \gamma \mathbb{E}_{\boldsymbol{y}_{\leq t+1} \sim T(\cdot | \boldsymbol{y}_{\leq t}, y_{t+1})} \left[ V^{\pi_*}(\boldsymbol{y}_{\leq t+1}) \right] \right) \right) \right]$$

$$= \mathbb{E}_{\substack{\boldsymbol{x} \sim p_{\boldsymbol{x}} \\ \tau \sim \pi_\theta | \boldsymbol{x}}} \left[ \sum_{t=0}^{T-1} \gamma^t \left( V^{\pi_*}(\boldsymbol{y}_{\leq t}) - Q^{\pi_*}(\boldsymbol{y}_{\leq t}, y_{t+1}) \right) \right]$$

$$= \mathbb{E}_{\substack{\boldsymbol{x} \sim p_{\boldsymbol{x}} \\ \tau \sim \pi_\theta | \boldsymbol{x}}} \left[ \sum_{t=0}^{T-1} \gamma^t \left( \mathbb{E}_{y \sim \pi_*(\cdot | \boldsymbol{y}_{\leq t})} \left[ Q^{\pi_*}(\boldsymbol{y}_{\leq t}, y) \right] - Q^{\pi_*}(\boldsymbol{y}_{\leq t+1}, y_{t+1}) \right) \right]$$

$$\leq \sup_{f \in \mathcal{F}_{Q_*}} \mathbb{E}_{\substack{\boldsymbol{x} \sim p_{\boldsymbol{x}} \\ \tau \sim \pi_\theta | \boldsymbol{x}}} \left[ \sum_{t=0}^{T-1} \gamma^t \left( \mathbb{E}_{y \sim \pi_*(\cdot | \boldsymbol{y}_{\leq t})} \left[ f(\boldsymbol{y}_{\leq t}, y) \right] - f(\boldsymbol{y}_{\leq t}, y_{t+1}) \right) \right],$$

which completes the proof of Proposition 2. $\qquad\square$

### A.3 Existing Step-Wise Distribution Distance for Distillation

**Definition 4** (**Step-wise distribution distances for distillation [39]**). *Following Wen et al. [39], we define four groups of well-studied probability distribution distances as follows,*

- **Total variation (TV) distance.** *The token-level TV distance between the probabilities of teacher policy $\pi_*$ and student policy $\pi_\theta$ given the current state $\boldsymbol{y}_{\leq t}$ can be defined by the $\ell_2$-norm as follows,*

$$\mathrm{TV}(\pi_\theta(\cdot|\boldsymbol{y}_{\leq t}), \pi_*(\cdot|\boldsymbol{y}_{\leq t})) := \frac{1}{2}\sum_{y\in\mathcal{V}}\left|\pi_*(y|\boldsymbol{y}_{\leq t}) - \pi_\theta(y|\boldsymbol{y}_{\leq t})\right| \tag{14}$$

- **Kullback–Leibler (KL) divergence.** *The token-level KL divergence between the probabilities of teacher policy $\pi_*$ and student policy $\pi_\theta$ given the current state $\boldsymbol{y}_{\leq t}$ can be defined as follows,*

$$\mathrm{KL}(\pi_\theta(\cdot|\boldsymbol{y}_{\leq t}), \pi_*(\cdot|\boldsymbol{y}_{\leq t})) := \sum_{y\in\mathcal{V}}\pi_*(y|\boldsymbol{y}_{\leq t})\log\frac{\pi_*(y|\boldsymbol{y}_{\leq t})}{\pi_\theta(y|\boldsymbol{y}_{\leq t})} \tag{15}$$

- **Reverse Kullback–Leibler (RKL) divergence.** *The token-level RKL divergence between the probabilities of teacher policy $\pi_*$ and student policy $\pi_\theta$ given the current state $boldsymbol{y}_{\leq t}$ can be defined as follows,*

$$\mathrm{RKL}(\pi_\theta(\cdot|\boldsymbol{y}_{\leq t}), \pi_*(\cdot|\boldsymbol{y}_{\leq t})) := \sum_{y\in\mathcal{V}}\pi_\theta(y|\boldsymbol{y}_{\leq t})\log\frac{\pi_\theta(y|\boldsymbol{y}_{\leq t})}{\pi_*(y|\boldsymbol{y}_{\leq t})} \tag{16}$$

- **Jenson–Shannon (JS) divergence.** *The token-level JS divergence between the probabilities of teacher policy $\pi_*$ and student policy $\pi_\theta$ given the current state $boldsymbol{y}_{\leq t}$ can be defined based on the KL divergence and RKL divergence as follows,*

$$\mathrm{JS}(\pi_\theta(\cdot|\boldsymbol{y}_{\leq t}), \pi_*(\cdot|\boldsymbol{y}_{\leq t})) := \frac{1}{2}\mathrm{KL}(\pi_*, \frac{\pi_\theta + \pi_*}{2}) + \frac{1}{2}\mathrm{RKL}(\pi_\theta, \frac{\pi_\theta + \pi_*}{2}) \tag{17}$$

### A.4 Proof of Theorem 1

*Proof.* We first derive an upper bound for the on-policy moment-matching objective of Eq. (3). Set $\mathcal{F}_{Q_*} = \{f : \|f\|_\infty \leq 1\}$, and by the definition of $\mathcal{L}(\pi_\theta, f)$ in Eq. (3), we have

$$\sup_{f:\|f\|_\infty\leq 1}\mathcal{L}^{\mathrm{on}}(\pi_\theta, f)$$

$$= \sup_{f:\|f\|_\infty\leq 1}\mathop{\mathbb{E}}_{\substack{\boldsymbol{x}\sim p_{\boldsymbol{x}}\\\tau\sim\pi_\theta|\boldsymbol{x}}}\left[\sum_{t=0}^{T-1}\gamma^t\left(\mathop{\mathbb{E}}_{y\sim\pi_*(\cdot|\boldsymbol{y}_{\leq t})}\big[f(\boldsymbol{y}_{\leq t}, y)\big] - f(\boldsymbol{y}_{\leq t}, y)\right)\right]$$

$$= \sup_{f:\|f\|_\infty\leq 1}\mathop{\mathbb{E}}_{\substack{\boldsymbol{x}\sim p_{\boldsymbol{x}}\\\tau\sim\pi_\theta|\boldsymbol{x}}}\left[\sum_{t=0}^{T-1}\gamma^t\left(\mathop{\mathbb{E}}_{y\sim\pi_*(\cdot|\boldsymbol{y}_{\leq t})}\big[f(\boldsymbol{y}_{\leq t}, y)\big] - \mathop{\mathbb{E}}_{y\sim\pi_\theta(\cdot|\boldsymbol{y}_{\leq t})}\big[f(\boldsymbol{y}_{\leq t}, y)\big]\right)\right]$$

Then, we have

$$\sup_{f:\|f\|_\infty\leq 1}\mathcal{L}^{\mathrm{on}}(\pi_\theta, f)$$

$$\overset{(i)}{\leq} \mathop{\mathbb{E}}_{\substack{\boldsymbol{x}\sim p_{\boldsymbol{x}}\\\tau\sim\pi_\theta|\boldsymbol{x}}}\left[\sum_{t=0}^{T-1}\gamma^t\sup_{f:\|f\|_\infty\leq 1}\left(\mathop{\mathbb{E}}_{y\sim\pi_*(\cdot|\boldsymbol{y}_{\leq t})}\big[f(\boldsymbol{y}_{\leq t}, y)\big] - \mathop{\mathbb{E}}_{y\sim\pi_\theta(\cdot|\boldsymbol{y}_{\leq t}))}\big[f(\boldsymbol{y}_{\leq t}, y)\big]\right)\right]$$

$$\overset{(ii)}{=} \mathop{\mathbb{E}}_{\substack{\boldsymbol{x}\sim p_{\boldsymbol{x}}\\\tau\sim\pi_\theta|\boldsymbol{x}}}\left[\sum_{t=0}^{T-1}\gamma^t\sum_{y\in\mathcal{V}}\big|\pi_*(y|\boldsymbol{y}_{\leq t}) - \pi_\theta(y|\boldsymbol{y}_{\leq t})\big|\right] \overset{(iii)}{=} 2d_{\mathrm{TV}}^{\mathrm{on}}(\pi_\theta, \pi_*) \text{ (Def. 2 \& 4)},$$

where $(i)$ follows from Jensen's inequality, $(ii)$ follows from [31] and $(iii)$ follows from the definition of TV distance.

Similarly, we can bound the off-policy version of Eq. (2) as follows,

$$\sup_{f:\|f\|_\infty \leq 1} \mathbb{E}_{\substack{\boldsymbol{x} \sim p_{\boldsymbol{x}} \\ \tau \sim \pi_* | \boldsymbol{x}}} \left[ \mathcal{L}^{\text{off}}(\pi_\theta, f) \right] \leq \mathbb{E}_{\substack{\boldsymbol{x} \sim p_{\boldsymbol{x}} \\ \tau \sim \pi_* | \boldsymbol{x}}} \left[ \sum_{t=0}^{T-1} \gamma^t \sum_{y \in \mathcal{V}} \left| \pi_*(y | \boldsymbol{y}_{\leq t}) - \pi_\theta(y | \boldsymbol{y}_{\leq t}) \right| \right]$$

$$= 2 d_{\text{TV}}^{\text{off}}(\pi_\theta, \pi_*) \text{ (Def. 4)},$$

which completes the proof of Theorem 1. $\qquad\qquad\square$

## A.5 Derivation of Policy Gradient in Eq. (10)

Based on the definition of training objective in Eq. (9), we have

$$\nabla \mathcal{L}(\pi_\theta, f_{\phi_1}, f_{\phi_2}) = \beta \nabla \mathcal{L}^{\text{off}}(\pi_\theta, f_{\phi_1}) + (1 - \beta) \nabla \mathcal{L}^{\text{on}}(\pi_\theta, f_{\phi_2}) \tag{18}$$

Based on the definition of $\mathcal{L}^{\text{off}}(\pi_\theta, f_{\phi_1})$ in Eq. (2), we have

$$\begin{aligned}
&\nabla \mathcal{L}^{\text{off}}(\pi_\theta, f_{\phi_1}) \\
&= \nabla \mathbb{E}_{\substack{\boldsymbol{x} \sim p_{\boldsymbol{x}} \\ \tau \sim \pi_* | \boldsymbol{x}}} \left[ \sum_{t=0}^{T-1} \gamma^t \left( f(\boldsymbol{y}_{\leq t}, y_{t+1}) - \mathbb{E}_{y \sim \pi_\theta(\cdot | \boldsymbol{y}_{\leq t})} \left[ f(\boldsymbol{y}_{\leq t}, y) \right] \right) \right] \\
&= - \mathbb{E}_{\substack{\boldsymbol{x} \sim p_{\boldsymbol{x}} \\ \tau \sim \pi_* | \boldsymbol{x}}} \sum_{t=0}^{T-1} \gamma^t \nabla \mathbb{E}_{y \sim \pi_\theta(\cdot | \boldsymbol{y}_{\leq t})} \left[ f_{\phi_1}(\boldsymbol{y}_{\leq t}, y) \right] \\
&= - \mathbb{E}_{\substack{\boldsymbol{x} \sim p_{\boldsymbol{x}} \\ \tau \sim \pi_* | \boldsymbol{x}}} \sum_{t=0}^{T-1} \gamma^t \sum_{y \in \mathcal{V}} \pi_\theta(y | \boldsymbol{y}_{\leq t}) \nabla \log \pi_\theta(y | \boldsymbol{y}_{\leq t}) f_{\phi_1}(\boldsymbol{y}_{\leq t}, y) \\
&= - \mathbb{E}_{\substack{\boldsymbol{x} \sim p_{\boldsymbol{x}} \\ \tau \sim \pi_* | \boldsymbol{x}}} \sum_{t=0}^{T-1} \gamma^t \mathbb{E}_{y \sim \pi_\theta(\cdot | \boldsymbol{y}_{\leq t})} \left[ \nabla \log \pi_\theta(y | \boldsymbol{y}_{\leq t}) f_{\phi_1}(\boldsymbol{y}_{\leq t}, y) \right]
\end{aligned} \tag{19}$$

Then, based on the definition of $\mathcal{L}^{\text{on}}(\pi_\theta, f_{\phi_2})$ in Eq. (3), we have

$$\begin{aligned}
&\nabla \mathcal{L}^{\text{on}}(\pi_\theta, f_{\phi_2}) \\
&= \nabla \mathbb{E}_{\substack{\boldsymbol{x} \sim p_{\boldsymbol{x}} \\ \tau \sim \pi_\theta | \boldsymbol{x}}} \left[ \sum_{t=0}^{T-1} \gamma^t \left( \mathbb{E}_{y \sim \pi_*(\cdot | \boldsymbol{y}_{\leq t})} \left[ f_{\phi_2}(\boldsymbol{y}_{\leq t}, y) \right] - f_{\phi_2}(\boldsymbol{y}_{\leq t}, y_{t+1}) \right) \right] \\
&\overset{(i)}{=} \mathbb{E}_{\substack{\boldsymbol{x} \sim p_{\boldsymbol{x}} \\ \tau \sim \pi_\theta | \boldsymbol{x}}} \left[ \sum_{t=0}^{T-1} \gamma^t \nabla \log \pi_\theta(y_t | \boldsymbol{y}_{\leq t}) \sum_{t'=t}^{T-1} \gamma^{t'-t} \left( \mathbb{E}_{y \sim \pi_*(\cdot | \boldsymbol{y}_{\leq t'})} \left[ f_{\phi_2}(\boldsymbol{y}_{\leq t'}, y) \right] - f_{\phi_2}(\boldsymbol{y}_{\leq t'}, y_{t'+1}) \right) \right],
\end{aligned} \tag{20}$$

where $(i)$ follows from a standard derivation of gradient policy (c.f. [22]). For simplicity, set

$$\hat{Q}_{f_{\phi_2}}(\boldsymbol{y}_{\leq t}, y_{t+1}) = \sum_{t'=t}^{T-1} \gamma^{t'-t} \left( \mathbb{E}_{y \sim \pi_*(\cdot | \boldsymbol{y}_{\leq t'})} \left[ f_{\phi_2}(\boldsymbol{y}_{\leq t'}, y) \right] - f_{\phi_2}(\boldsymbol{y}_{\leq t'}, y_{t'+1}) \right) \tag{21}$$

as the empirical $Q$-value given any draw of trajectory $\tau \sim \pi_\theta | \boldsymbol{y}_0 = \boldsymbol{x}, \boldsymbol{x} \sim p_{\boldsymbol{x}}$ in Eq. (20).

Coming back to Eq. (18) and combining with Eq. (19) and Eq. (20), we have

$$\begin{aligned}
\nabla \mathcal{L}(\pi_\theta, f_{\phi_1}, f_{\phi_2}) = &- \beta \mathbb{E}_{\substack{\boldsymbol{x} \sim p_{\boldsymbol{x}} \\ \tau \sim \pi_* | \boldsymbol{x}}} \left[ \sum_{t=0}^{T-1} \gamma^t \mathbb{E}_{y \sim \pi_\theta(\cdot | \boldsymbol{y}_{\leq t})} \left[ \nabla \log \pi_\theta(y | \boldsymbol{y}_{\leq t}) f_{\phi_1}(\boldsymbol{y}_{\leq t}, y) \right] \right] \\
&+ (1 - \beta) \mathbb{E}_{\substack{\boldsymbol{x} \sim p_{\boldsymbol{x}} \\ \tau \sim \pi_\theta | \boldsymbol{x}}} \left[ \sum_{t=0}^{T-1} \gamma^t \nabla \log \pi_\theta(y_{t+1} | \boldsymbol{y}_{\leq t}) \hat{Q}_{f_{\phi_2}}(\boldsymbol{y}_{\leq t}, y_{t+1}) \right]
\end{aligned}$$

Then, using the law of iterated expectations, we obtain the final formulation of policy gradient,

$$\nabla \mathcal{L}(\pi_\theta, f_{\phi_1}, f_{\phi_2}) = \mathbb{E}_{\boldsymbol{x} \sim p_{\boldsymbol{x}}} \left[ -\beta \mathbb{E}_{\tau \sim \pi_* | \boldsymbol{x}} \left[ \sum_{t=0}^{T-1} \gamma^t \mathbb{E}_{y \sim \pi_\theta(\cdot | \boldsymbol{y}_{\leq t})} \left[ \nabla \log \pi_\theta(y | \boldsymbol{y}_{\leq t}) f_{\phi_1}(\boldsymbol{y}_{\leq t}, y) \right] \right] \right.$$

$$\left. + (1-\beta) \mathbb{E}_{\tau' \sim \pi_\theta | \boldsymbol{x}} \left[ \sum_{t=0}^{T-1} \gamma^t \nabla \log \pi_\theta(y'_{t+1} | \boldsymbol{y}'_{\leq t}) \hat{Q}_{f_{\phi_2}}(\boldsymbol{y}'_{\leq t}, y'_{t+1}) \right] \right],$$

which completes the derivation of policy gradient in Eq. (10).

## A.6 Proof of Theorem 2

**Lemma 1.** *Let $\hat{\nabla} \mathcal{L}(\theta) = -\beta \hat{\mathcal{G}}^{\mathrm{off}}(\tau, \theta) + (1-\beta) \hat{\mathcal{G}}^{\mathrm{on}}(\tau', \theta)$ denote the empirical policy gradient given any trajectories $\boldsymbol{x} \sim p_{\boldsymbol{x}}, \tau \sim \pi_* | \boldsymbol{x}, \tau' \sim \pi_\theta | \boldsymbol{x}$, where $\mathcal{L}(\theta) := \mathcal{L}(\pi_\theta, f_{\phi_1}, f_{\phi_2})$ (def. in Eq. (9)) denote the objective w.r.t. the policy parameters $\theta$ given any off-/on-policy Q-value functions $f_{\phi_1}$ and $f_{\phi_2}$. Then, under Assumption 1, we have $\|\hat{\nabla} \mathcal{L}(\theta)\| \leq B_{\mathcal{L}}$ with*

$$B_{\mathcal{L}} = \frac{\beta(1-\gamma^T)B}{1-\gamma} + \frac{2(1-\beta)(1-\gamma^T)^2 B}{(1-\gamma)^2}$$

*Proof.* By triangle inequality, we have for any $\boldsymbol{x} \sim p_{\boldsymbol{x}}, \tau \sim \pi_* | \boldsymbol{x}, \tau' \sim \pi_\theta | \boldsymbol{x}$,

$$\|\hat{\nabla} \mathcal{L}(\theta)\| \leq \beta \left\| \hat{\mathcal{G}}^{\mathrm{off}}(\tau, \theta) \right\| + (1-\beta) \left\| \hat{\mathcal{G}}^{\mathrm{on}}(\tau', \theta) \right\| \tag{22}$$

By the formulation of off-policy gradient $\|\hat{\mathcal{G}}^{\mathrm{off}}(\tau, \theta)\|$ in Eq. (10) under the condition of optimized off-policy Q-value functions $f_{\phi_1}$ by Algorithm 1, we have

$$\left\| \hat{\mathcal{G}}^{\mathrm{off}}(\tau, \theta) \right\| = \left\| \sum_{t=0}^{T-1} \gamma^t \mathbb{E}_{y \sim \pi_\theta(\cdot | \boldsymbol{y}_{\leq t})} \left[ \nabla \log \pi_\theta(y | \boldsymbol{y}_{\leq t}) f_{\phi_1}(\boldsymbol{y}_{\leq t}, y) \right] \right\|$$

By Jensen's inequality, we have

$$\left\| \hat{\mathcal{G}}^{\mathrm{off}}(\tau, \theta) \right\| \leq \sum_{t=0}^{T-1} \gamma^t \mathbb{E}_{y \sim \pi_\theta(\cdot | \boldsymbol{y}_{\leq t})} \left[ \left\| \nabla \log \pi_\theta(y | \boldsymbol{y}_{\leq t}) \right\| \left| f_{\phi_1}(\boldsymbol{y}_{\leq t}, y) \right| \right]$$

By Assumption 1, we have

$$\left\| \hat{\mathcal{G}}^{\mathrm{off}}(\tau, \theta) \right\| \leq B \sum_{t=0}^{T-1} \gamma^t = \frac{B(1-\gamma^T)}{1-\gamma} \tag{23}$$

Similarly, we can bound the on-policy gradient $\|\hat{\mathcal{G}}^{\mathrm{on}}(\tau', \theta)\|$ by Jensen's inequality as follows,

$$\left\| \hat{\mathcal{G}}^{\mathrm{on}}(\tau', \theta) \right\| \leq \sum_{t=0}^{T-1} \gamma^t \left\| \nabla \log \pi_\theta(y_{t+1} | \boldsymbol{y}_{\leq t}) \right\| \left| \hat{Q}_{f_{\phi_2}}(\boldsymbol{y}_{\leq t}, y_{t+1}) \right|$$

Based on the definition of $\hat{Q}_{f_{\phi_2}}(\boldsymbol{y}_{\leq t}, y_{t+1})$ in Eq. (21) and by Jensen's inequality, we have

$$\left| \hat{Q}_{f_{\phi_2}}(\boldsymbol{y}_{\leq t}, y_{t+1}) \right| = \left| \sum_{t'=t}^{T-1} \gamma^{t'-t} \left( \mathbb{E}_{y \sim \pi_*(\cdot | \boldsymbol{y}_{\leq t'})} \left[ f_{\phi_2}(\boldsymbol{y}_{\leq t'}, y) \right] - f_{\phi_2}(\boldsymbol{y}_{\leq t'}, y_{t'+1}) \right) \right|$$

$$\leq \sum_{t'=t}^{T-1} \gamma^{t'-t} \left( \mathbb{E}_{y \sim \pi_*(\cdot | \boldsymbol{y}_{\leq t'})} \left[ \left| f_{\phi_2}(\boldsymbol{y}_{\leq t'}, y) \right| \right] + \left| f_{\phi_2}(\boldsymbol{y}_{\leq t'}, y_{t'+1}) \right| \right)$$

Then, by Assumption 1 (i) that $\|f_{\phi_2}\|_\infty \leq 1$, we have

$$\left| \hat{Q}_{f_{\phi_2}}(\boldsymbol{y}_{\leq t}, y_{t+1}) \right| \leq 2 \sum_{t'=t}^{T-1} \gamma^{t'-t} \leq 2 \sum_{t'=0}^{T-1} \gamma^{t'} = \frac{2(1-\gamma^T)}{1-\gamma}$$

Thus, we have

$$\left\|\hat{\mathcal{G}}^{\mathrm{on}}(\tau', \theta)\right\| \leq \frac{2(1 - \gamma^T)^2 B}{(1 - \gamma)^2} \tag{24}$$

Coming back to the bound of $\|\hat{\nabla}\mathcal{L}(\theta)\|$ in Eq. (22), we combine it with Eq. (23) and Eq. (24). Then, we have

$$\|\hat{\nabla}\mathcal{L}(\theta)\| \leq \underbrace{\frac{\beta(1 - \gamma^T)B}{1 - \gamma} + \frac{2(1 - \beta)(1 - \gamma^T)^2 B}{(1 - \gamma)^2}}_{B_{\mathcal{L}}},$$

which completes the proof of Lemma 1. $\qquad\square$

**Lemma 2.** *Under Assumption 1, the objective function $\mathcal{L}(\theta)$ is $L_{\mathcal{L}}$-smooth such that for any $\theta, \theta' \in \Theta$,*

$$\mathcal{L}(\theta) \leq \mathcal{L}(\theta') + \langle \nabla\mathcal{L}(\theta'), \theta - \theta'\rangle + \frac{1}{2}L_{\mathcal{L}}\|\theta - \theta'\|^2,$$

*with the constant*

$$L_{\mathcal{L}} = \beta\frac{(1 - \gamma^T)(B^2 + L)}{1 - \gamma} + (1 - \beta)\frac{2(1 - \gamma^T)^2}{(1 - \gamma)^2}\left(\frac{\gamma B^2}{1 - \gamma} + L\right)$$

*Proof.* Under the definition of policy gradient in Eq. (10), for any $\theta_1, \theta_2 \in \Theta$, we have

$$\|\nabla\mathcal{L}(\theta_1) - \nabla\mathcal{L}(\theta_2)\|$$
$$= \left\| \mathbb{E}_{\boldsymbol{x}\sim p_{\boldsymbol{x}}}\left[ -\beta \mathbb{E}_{\tau\sim\pi_*|\boldsymbol{x}}\left[\hat{\mathcal{G}}^{\mathrm{off}}(\tau, \theta_1) - \hat{\mathcal{G}}^{\mathrm{off}}(\tau, \theta_2)\right] \right.\right.$$
$$\left.\left. + (1 - \beta)\left(\mathbb{E}_{\tau_1\sim\pi_{\theta_1}|\boldsymbol{x}}\left[\hat{\mathcal{G}}^{\mathrm{on}}(\tau_1, \theta_1)\right] - \mathbb{E}_{\tau_2\sim\pi_{\theta_2}|\boldsymbol{x}}\left[\hat{\mathcal{G}}^{\mathrm{on}}(\tau_2, \theta_2)\right]\right) \right] \right\|$$

Then, by Jensen's inequality and triangle inequality, we have

$$\|\nabla\mathcal{L}(\theta_1) - \nabla\mathcal{L}(\theta_2)\|$$
$$\leq \mathbb{E}_{\boldsymbol{x}\sim p_{\boldsymbol{x}}}\left[ \beta \mathbb{E}_{\tau\sim\pi_*|\boldsymbol{x}}\underbrace{\left\|\hat{\mathcal{G}}^{\mathrm{off}}(\tau, \theta_1) - \hat{\mathcal{G}}^{\mathrm{off}}(\tau, \theta_2)\right\|}_{I_1} \right. \tag{25}$$
$$\left. + (1 - \beta)\underbrace{\left\|\mathbb{E}_{\tau_1\sim\pi_{\theta_1}|\boldsymbol{x}}\left[\hat{\mathcal{G}}^{\mathrm{on}}(\tau_1, \theta_1)\right] - \mathbb{E}_{\tau_2\sim\pi_{\theta_2}|\boldsymbol{x}}\left[\hat{\mathcal{G}}^{\mathrm{on}}(\tau_2, \theta_2)\right]\right\|}_{I_2} \right]$$

Based on the definition of off-policy gradient in Eq. (10) and using Jensen's inequality, we have for any $\boldsymbol{x}\sim p_{\boldsymbol{x}}, \tau\sim\pi_*|\boldsymbol{x}$,

$$I_1 = \left\|\hat{\mathcal{G}}^{\mathrm{off}}(\tau, \theta_1) - \hat{\mathcal{G}}^{\mathrm{off}}(\tau, \theta_2)\right\|$$
$$\leq \sum_{t=0}^{T-1}\gamma^t\left\| \mathbb{E}_{y\sim\pi_{\theta_1}(\cdot|\boldsymbol{y}_{\leq t})}\left[\nabla\log\pi_{\theta_1}(y|\boldsymbol{y}_{\leq t})f_{\phi_1}(\boldsymbol{y}_{\leq t}, y)\right] - \mathbb{E}_{y\sim\pi_{\theta_2}(\cdot|\boldsymbol{y}_{\leq t})}\left[\nabla\log\pi_{\theta_2}(y|\boldsymbol{y}_{\leq t})f_{\phi_1}(\boldsymbol{y}_{\leq t}, y)\right]\right\|$$
$$\tag{26}$$

Then, by triangle inequality, we have for any $t \in \{0, 1, \ldots, T-1\}$,

$$
\left\| \mathop{\mathbb{E}}_{y \sim \pi_{\theta_1}(\cdot|\boldsymbol{y}_{\leq t})} \left[ \nabla \log \pi_{\theta_1}(y|\boldsymbol{y}_{\leq t}) f_{\phi_1}(\boldsymbol{y}_{\leq t}, y) \right] - \mathop{\mathbb{E}}_{y \sim \pi_{\theta_2}(\cdot|\boldsymbol{y}_{\leq t})} \left[ \nabla \log \pi_{\theta_2}(y|\boldsymbol{y}_{\leq t}) f_{\phi_1}(\boldsymbol{y}_{\leq t}, y) \right] \right\|
$$

$$
= \left\| \sum_{y \in \mathcal{V}} \left[ \pi_{\theta_1}(y|\boldsymbol{y}_{\leq t}) \nabla \log \pi_{\theta_1}(y|\boldsymbol{y}_{\leq t}) f_{\phi_1}(\boldsymbol{y}_{\leq t}, y) - \pi_{\theta_2}(y|\boldsymbol{y}_{\leq t}) \nabla \log \pi_{\theta_2}(y|\boldsymbol{y}_{\leq t}) f_{\phi_1}(\boldsymbol{y}_{\leq t}, y) \right] \right\|
$$

$$
\leq \sum_{y \in \mathcal{V}} \left| f_{\phi_1}(\boldsymbol{y}_{\leq t}, y) \right| \left[ \left| \pi_{\theta_1}(y|\boldsymbol{y}_{\leq t}) - \pi_{\theta_2}(y|\boldsymbol{y}_{\leq t}) \right| \left\| \nabla \log \pi_{\theta_1}(y|\boldsymbol{y}_{\leq t}) \right\| \right.
$$

$$
\left. + \pi_{\theta_2}(y|\boldsymbol{y}_{\leq t}) \left\| \nabla \log \pi_{\theta_1}(y|\boldsymbol{y}_{\leq t}) - \nabla \log \pi_{\theta_2}(y|\boldsymbol{y}_{\leq t}) \right\| \right]
$$

$$(27)$$

By Taylor expansion of $\pi_\theta(y|\boldsymbol{y}_{\leq t})$, we have that for any $t \in \{0, 1, \ldots, T-1\}$,

$$
\left| \pi_{\theta_1}(y|\boldsymbol{y}_{\leq t}) - \pi_{\theta_2}(y|\boldsymbol{y}_{\leq t}) \right| = \left| (\theta_1 - \theta_2)^\top \nabla \log \pi_{\tilde{\theta}}(y|\boldsymbol{y}_{\leq t}) \pi_{\tilde{\theta}}(y|\boldsymbol{y}_{\leq t}) \right|
$$

$$
\leq \|\theta_1 - \theta_2\| \|\nabla \log \pi_{\tilde{\theta}}(y|\boldsymbol{y}_{\leq t})\| \pi_{\tilde{\theta}}(y|\boldsymbol{y}_{\leq t})
$$

$$
\leq \|\theta_1 - \theta_2\| \cdot B \cdot \pi_{\tilde{\theta}}(y|\boldsymbol{y}_{\leq t}),
$$

where $\tilde{\theta}$ is a vector lying between $\theta_1$ and $\theta_2$, i.e., there exists some $\lambda \in [0, 1]$ such that $\tilde{\theta} = \lambda \theta_1 + (1-\lambda)\theta_2$. Then, combining with Eq. (27), yields

$$
\left\| \mathop{\mathbb{E}}_{y \sim \pi_{\theta_1}(\cdot|\boldsymbol{y}_{\leq t})} \left[ \nabla \log \pi_{\theta_1}(y|\boldsymbol{y}_{\leq t}) f_{\phi_1}(\boldsymbol{y}_{\leq t}, y) \right] - \mathop{\mathbb{E}}_{y \sim \pi_{\theta_2}(\cdot|\boldsymbol{y}_{\leq t})} \left[ \nabla \log \pi_{\theta_2}(y|\boldsymbol{y}_{\leq t}) f_{\phi_1}(\boldsymbol{y}_{\leq t}, y) \right] \right\|
$$

$$
\leq \sum_{y \in \mathcal{V}} \left[ B^2 \pi_{\tilde{\theta}}(y|\boldsymbol{y}_{\leq t}) \|\theta_1 - \theta_2\| + \pi_{\theta_2}(y|\boldsymbol{y}_{\leq t}) L \|\theta_1 - \theta_2\| \right]
$$

$$
= (B^2 + L)\|\theta_1 - \theta_2\|
$$

Then, combining with Eq. (26) yields

$$
I_1 = \left\| \hat{\mathcal{G}}^{\mathrm{off}}(\tau, \theta_1) - \hat{\mathcal{G}}^{\mathrm{off}}(\tau, \theta_2) \right\| \leq (B^2 + L)\|\theta_1 - \theta_2\| \sum_{t=0}^{T-1} \gamma^t \leq \frac{(1-\gamma^T)(B^2+L)}{1-\gamma}\|\theta_1 - \theta_2\|
$$

$$(28)$$

In addition, we can first bound $I_2$ using Jensen's inequality and triangle inequality,

$$
I_2 = \left\| \mathbb{E}_{\tau_1 \sim \pi_{\theta_1}|\boldsymbol{x}}\left[ \hat{\mathcal{G}}^{\mathrm{on}}(\tau_1, \theta_1) \right] - \mathbb{E}_{\tau_2 \sim \pi_{\theta_2}|\boldsymbol{x}}\left[ \hat{\mathcal{G}}^{\mathrm{on}}(\tau_2, \theta_2) \right] \right\|
$$

$$
\leq \sum_{t=0}^{T-1} \int \gamma^t |\hat{Q}_{f_{\phi_2}}(\boldsymbol{y}_{\leq t}, y_{t+1})| \left| \prod_{t'=0}^{t-1} \pi_{\theta_1}(y_{t'+1}|\boldsymbol{y}_{\leq t'}) \nabla \log \pi_{\theta_1}(y_{t'+1}|\boldsymbol{y}_{\leq t'}) \right.
$$

$$
\left. - \prod_{t'=0}^{t-1} \pi_{\theta_2}(y_{t'+1}|\boldsymbol{y}_{\leq t'}) \nabla \log \pi_{\theta_2}(y_{t'+1}|\boldsymbol{y}_{\leq t'}) \right| d\boldsymbol{y}_{\leq 1} \cdots d\boldsymbol{y}_{\leq t} dy_1 \cdots dy_t
$$

By triangle inequality and the boundess of $|\hat{Q}_{f_{\phi_2}}(\boldsymbol{y}_{\leq t}, y_{t+1})| \leq \frac{2(1-\gamma^T)}{1-\gamma}$, we further have,

$$
I_2 \leq \frac{2(1-\gamma^T)}{1-\gamma} \sum_{t=0}^{T-1} \int \gamma^t \left( \left| \prod_{t'=0}^{t-1} \pi_{\theta_1}(y_{t'+1}|\boldsymbol{y}_{\leq t'}) - \prod_{t'=0}^{t-1} \pi_{\theta_2}(y_{t'+1}|\boldsymbol{y}_{\leq t'}) \right| \|\nabla \log \pi_{\theta_1}(y_{t'+1}|\boldsymbol{y}_{\leq t'})\| \right.
$$

$$
\left. + \prod_{t'=0}^{t-1} \pi_{\theta_2}(y_{t'+1}|\boldsymbol{y}_{\leq t'}) \|\nabla \log \pi_{\theta_1}(y_{t'+1}|\boldsymbol{y}_{\leq t'}) - \nabla \log \pi_{\theta_2}(y_{t'+1}|\boldsymbol{y}_{\leq t'})\| \right) d\boldsymbol{y}_{\leq 1} \cdots d\boldsymbol{y}_{\leq t} dy_1 \cdots dy_t
$$

$$(29)$$

By Taylor expansion of $\prod_{t'=0}^{t-1} \pi_\theta(y_{t'+1}|\boldsymbol{y}_{\leq t'})$, we have

$$
\left| \prod_{t'=0}^{t-1} \pi_{\theta_1}(y_{t'+1}|\boldsymbol{y}_{\leq t'}) - \prod_{t'=0}^{t-1} \pi_{\theta_2}(y_{t'+1}|\boldsymbol{y}_{\leq t'}) \right|
$$

$$
= \left| (\theta_1 - \theta_2)^\top \sum_{t'=0}^{t-1} \nabla \pi_{\tilde{\theta}}(y_{t'+1}|\boldsymbol{y}_{\leq t'}) \prod_{t''=0,t''\neq t'}^{t-1} \pi_{\tilde{\theta}}(y_{t''+1}|\boldsymbol{y}_{\leq t''}) \right|
$$

$$
\leq \|\theta_1 - \theta_2\| \sum_{t'=0}^{t-1} \|\nabla \log \pi_{\tilde{\theta}}(y_{t'+1}|\boldsymbol{y}_{\leq t'})\| \prod_{t''=0}^{t-1} \pi_{\tilde{\theta}}(y_{t''+1}|\boldsymbol{y}_{\leq t''})
$$

$$
\leq \|\theta_1 - \theta_2\| \cdot t \cdot B \cdot \prod_{t''=0}^{t-1} \pi_{\tilde{\theta}}(y_{t''+1}|\boldsymbol{y}_{\leq t''}),
$$

where $\tilde{\theta}$ denotes a vector lying between $\theta_1$ and $\theta_2$, i.e., there exists some $\lambda$ such that $\tilde{\theta} = \lambda\theta_1 + (1 - \lambda)\theta_2$. Coming back to the boundness of $I_2$ in Eq. (29), we have

$$
I_2 \leq \frac{2(1-\gamma^T)}{1-\gamma} \sum_{t=0}^{T-1} \int \gamma^t \left( B^2 t \prod_{t''=0}^{t-1} \pi_{\tilde{\theta}}(y_{t''+1}|\boldsymbol{y}_{\leq t''}) + L \prod_{t'=0}^{t-1} \pi_{\theta_2}(y_{t'+1}|\boldsymbol{y}_{\leq t'}) \right) \|\theta_1 - \theta_2\| d\boldsymbol{y}_{\leq 1} \cdots d\boldsymbol{y}_{\leq t} dy_1 \cdots dy_t
$$

$$
= \frac{2(1-\gamma^T)}{1-\gamma} \sum_{t=0}^{T-1} \gamma^t (B^2 t + L) \|\theta_1 - \theta_2\| \leq \frac{2(1-\gamma^T)^2}{(1-\gamma)^2} \left( \frac{\gamma B^2}{1-\gamma} + L \right) \|\theta_1 - \theta_2\|,
$$

$$(30)$$

where the last inequality follows from the fact that

$$
\sum_{t=0}^{T-1} t\gamma^t = \frac{\gamma - T\gamma^T + (T-1)\gamma^{T+1}}{(1-\gamma)^2} \leq \frac{\gamma - T\gamma^{T+1} + (T-1)\gamma^{T+1}}{(1-\gamma)^2} = \frac{\gamma(1-\gamma^T)}{(1-\gamma)^2}
$$

Then, combining Eq. (25) with the boundness of $I_1$ in Eq. (28) and the boundness of $I_2$ in Eq. (30), we obtain the final bound of

$$
\|\nabla\mathcal{L}(\theta_1) - \nabla\mathcal{L}(\theta_2)\|
$$

$$
\leq \underbrace{\left( \beta \frac{(1-\gamma^T)(B^2 + L)}{1-\gamma} + (1-\beta)\frac{2(1-\gamma^T)^2}{(1-\gamma)^2} \left( \frac{\gamma B^2}{1-\gamma} + L \right) \right)}_{L_\mathcal{L}} \|\theta_1 - \theta_2\| \qquad (31)
$$

Next, we have for any $\theta, \theta \in \Theta$,

$$
\mathcal{L}(\theta) - \mathcal{L}(\theta') - \langle \nabla\mathcal{L}(\theta'), \theta - \theta' \rangle
$$

$$
\leq |\mathcal{L}(\theta) - \mathcal{L}(\theta') - \langle \nabla\mathcal{L}(\theta'), \theta - \theta' \rangle|
$$

$$
\leq \left| \int_{(0,1)} \langle \nabla\mathcal{L}(\theta' + t(\theta - \theta')), \theta - \theta' \rangle dt - \langle \nabla\mathcal{L}(\theta'), \theta - \theta' \rangle \right|
$$

$$
\leq \int_{(0,1)} \|\nabla\mathcal{L}(\theta' + t(\theta - \theta')) - \nabla\mathcal{L}(\theta')\| \|\theta - \theta'\| dt
$$

Then, by Eq. (31) and set $\theta_1 = \theta' + t(\theta - \theta')$ and $\theta_2 = \theta'$, we have

$$
\mathcal{L}(\theta) - \mathcal{L}(\theta') - \langle \nabla\mathcal{L}(\theta'), \theta - \theta' \rangle
$$

$$
\leq \int_{(0,1)} L_\mathcal{L} \|\theta - \theta'\|^2 t dt = \frac{1}{2} L_\mathcal{L} \|\theta - \theta'\|^2,
$$

which completes the proof of Lemma 2.

$$\square$$

We prove Theorem 2 as follows.

*Proof of Theorem 2.* Let $\theta_t$, $\theta_{t+1}$, $t \in \{0, 1, \ldots, T-1\}$ be adjacent parameters of policy $\pi_{\theta_t}$, $\pi_{\theta_{t+1}}$ given by Algorithm 1. Then, using Lemma 2 by setting $\theta = \theta_{k+1}, \theta' = \theta_k$ for any $k \in \{0, 1, \ldots, K-1\}$, we have

$$\mathcal{L}(\pi_{\theta_{k+1}}, f_{\phi_1(\theta_{k+1})}, f_{\phi_2(\theta_{k+1})})$$

$$\leq \mathcal{L}(\pi_{\theta_k}, f_{\phi_1(\theta_k)}, f_{\phi_2(\theta_k)}) + \langle \nabla \mathcal{L}(\pi_{\theta_k}, f_{\phi_1(\theta_k)}, f_{\phi_2(\theta_k)}), \theta_{k+1} - \theta_k \rangle + \frac{1}{2} L_{\mathcal{L}} \|\theta_{k+1} - \theta_k\|^2$$

Following from the updating rule $\theta_{k+1} = \theta_k - \eta \hat{\nabla} \mathcal{L}(\theta_k)$ and Lemma 1, we have

$$\|\theta_{k+1} - \theta_k\| = \eta \|\hat{\nabla} \mathcal{L}(\theta_k)\| \leq \eta B_{\mathcal{L}}$$

Then, we have

$$\mathcal{L}(\pi_{\theta_{k+1}}, f_{\phi_1(\theta_{k+1})}, f_{\phi_2(\theta_{k+1})})$$

$$\leq \mathcal{L}(\pi_{\theta_k}, f_{\phi_1(\theta_k)}, f_{\phi_2(\theta_k)}) - \langle \nabla \mathcal{L}(\pi_{\theta_k}, f_{\phi_1(\theta_k)}, f_{\phi_2(\theta_k)}), \eta \hat{\nabla} \mathcal{L}(\theta_k) \rangle + \frac{1}{2} \eta^2 B_{\mathcal{L}}^2 L_{\mathcal{L}}$$

We introduce a probability measure space $(\Omega, \mathcal{F}, P)$ and then $\theta_k : \Omega \to \Theta$, $k \in \{0, 1, \ldots, K-1\}$ can be viewed as a random variable on it. Let $\{\sigma(\theta_k)\}_{0 \leq k \leq K-1}$ denote a sequence of increasing sigma-algebras such that $\sigma(\theta_0) \subset \sigma(\theta_1) \subset \cdots \sigma(\theta_{K-1}) \subset \mathcal{F}$, we define the conditional expectation $\mathbb{E}[\hat{\nabla} \mathcal{L}(\theta_k) \mid \sigma(\theta_k)]$ as

$$\mathbb{E}[\hat{\nabla} \mathcal{L}(\theta_k) \mid \sigma(\theta_k)] = \mathbb{E}_{\boldsymbol{x} \sim p_{\boldsymbol{x}}} \left[ -\beta \mathbb{E}_{\tau \sim \pi_* | \boldsymbol{x}} \hat{\mathcal{G}}^{\text{off}}(\tau, \theta_k) + (1-\beta) \mathbb{E}_{\tau' \sim \pi_{\theta_k} | \boldsymbol{x}} \hat{\mathcal{G}}^{\text{on}}(\tau', \theta_k) \right]$$

$$= \nabla \mathcal{L}(\pi_{\theta_k}, f_{\phi_1(\theta_k)}, f_{\phi_2(\theta_k)}),$$

where the second equality follows from the unbiased estimation property in Eq. (10). Then, taking the conditional expectation, we have

$$\mathbb{E}[\mathcal{L}(\pi_{\theta_{k+1}}, f_{\phi_1(\theta_{k+1})}, f_{\phi_2(\theta_{k+1})}) | \sigma(\theta_k)]$$

$$\leq \mathcal{L}(\pi_{\theta_k}, f_{\phi_1(\theta_k)}, f_{\phi_2(\theta_k)}) - \eta \|\nabla \mathcal{L}(\pi_{\theta_k}, f_{\phi_1(\theta_k)}, f_{\phi_2(\theta_k)})\|^2 + \frac{1}{2} \eta^2 B_{\mathcal{L}}^2 L_{\mathcal{L}}$$

Taking total expectation, rearranging the terms and making average on $k \in \{0, 1, \ldots, K-1\}$, we have

$$\frac{1}{K} \sum_{k=0}^{K-1} \mathbb{E} \left[ \|\nabla \mathcal{L}(\pi_{\theta_k}, f_{\phi_1(\theta_k)}, f_{\phi_2(\theta_k)})\|^2 \right]$$

$$\leq \frac{1}{\eta K} \sum_{k=0}^{K-1} \left( \mathbb{E}[\mathcal{L}(\pi_{\theta_k}, f_{\phi_1(\theta_k)}, f_{\phi_2(\theta_k)})] - \mathbb{E}[\mathcal{L}(\pi_{\theta_{k+1}}, f_{\phi_1(\theta_{k+1})}, f_{\phi_2(\theta_{k+1})})] \right) + \frac{1}{2} \eta B_{\mathcal{L}}^2 L_{\mathcal{L}}$$

$$= \frac{1}{\eta K} \left( \mathcal{L}(\pi_{\theta_0}, f_{\phi_1(\theta_0)}, f_{\phi_2(\theta_0)}) - \mathbb{E}[\mathcal{L}(\pi_{\theta_K}, f_{\phi_1(\theta_K)}, f_{\phi_2(\theta_K)})] \right) + \frac{1}{2} \eta B_{\mathcal{L}}^2 L_{\mathcal{L}}$$

$$\leq \frac{2(1-\gamma^T)}{\eta(1-\gamma)K} + \frac{1}{2} \eta B_{\mathcal{L}}^2 L_{\mathcal{L}}$$

Let $\eta = \frac{2}{B_{\mathcal{L}}} \sqrt{\frac{1-\gamma^T}{(1-\gamma)K L_{\mathcal{L}}}}$ and denote $\mathcal{L}(\theta_k) = \mathcal{L}(\pi_{\theta_k}, f_{\phi_1(\theta_k)}, f_{\phi_2(\theta_k)})$ for simplicity for any $k \in \{0, 1, \ldots, K-1\}$, then we have

$$\min_{0 \leq k \leq K-1} \mathbb{E} \left[ \|\nabla \mathcal{L}(\theta_k)\|^2 \right] \leq \frac{1}{K} \sum_{k=0}^{K-1} \mathbb{E} \left[ \|\nabla \mathcal{L}(\theta_k)\|^2 \right] \leq 2 B_{\mathcal{L}} \sqrt{\frac{(1-\gamma^T) L_{\mathcal{L}}}{(1-\gamma)K}} = \mathcal{O}\left( \sqrt{\frac{1}{K}} \right) \quad (32)$$

$\square$

## A.7 Proof of Corollary 1

*Proof.* Let the convergence rate in Eq. (32) satisfy that $2B_{\mathcal{L}}\sqrt{\frac{(1-\gamma^T)L_{\mathcal{L}}}{(1-\gamma)K}} \leq \epsilon$, then we have

$$K \geq \frac{4(1-\gamma^T)B_{\mathcal{L}}^2 L_{\mathcal{L}}}{(1-\gamma)\epsilon^2},$$

which indicates that when the iteration number of policy updating satisfies $K := \mathcal{O}(\epsilon^{-2})$, it can reach an $\epsilon$-accurate stationary point to optimize the objective in Eq. (9), such that

$$\min_{0 \leq k \leq K-1} \mathbb{E}\left[\|\nabla\mathcal{L}(\theta_k)\|^2\right] \leq \epsilon$$

For simplicity, we define the policy as a softmax function with a linear transformation of $\boldsymbol{y}_{\leq t} \in \mathbb{R}^H$ with $\theta \in \mathbb{R}^{|\mathcal{V}| \times H}$. Formally, for any trajectory $\tau$ and any timestep $t \in \{0, 1, \ldots, T-1\}$, we have the probability of any $y \in \mathcal{V}$,

$$\pi_\theta(y|\boldsymbol{y}_{\leq t}) = \frac{\exp(\theta_y \boldsymbol{y}_{\leq t})}{\sum_{y' \in \mathcal{V}} \exp(\theta_{y'} \boldsymbol{y}_{\leq t})} \tag{33}$$

In the following, we will analyze the computational complexity in each policy updating iteration. First, we find that each inner-loop step of $Q$-value function updating has a gradient computation complexity of $\mathcal{O}(T|\mathcal{V}|H)$ given the linear formulation of $Q$-value functions. Accordingly, $N$ inner-loop steps in each policy updating iteration have a computational complexity of $\mathcal{O}(NT|\mathcal{V}|H)$. Second, for policy gradient computation, since computational complexity of $\nabla \log \pi_\theta(y|\boldsymbol{y}_{<T})$ is $\mathcal{O}(|\mathcal{V}|H)$, the computation complexity of policy gradient computation is $\mathcal{O}(T|\mathcal{V}|H(T + |\mathcal{V}|))$. Overall, the total gradient computational complexity is $\mathcal{O}(\epsilon^{-2}T|\mathcal{V}|H(N + T + |\mathcal{V}|))$, which completes the proof of Corollary 1.

$\square$

# B  Experimental Setup

We use NVIDIA A40 GPUs with 40GB RAM to conduct all the experiments.

## B.1  Instruction-Following Experiments

**Base models.** We conduct experiments on both GPT-2 [28] and OpenLLaMA [10]. For the GPT-2 experiments, we use GPT-2 XL[2] with 1.5B parameters to construct the teacher policy and GPT-2[3] with 117M parameters to construct the student policy. For the OpenLLaMA experiments, we use OpenLLaMA-7B[4] with 6.7B parameters to construct the teacher policy and OpenLLaMA-3B[5] with 2.7B parameters to construct the student model.

**Training details.** We fine-tune the OpenLLaMA-7B teacher model and the OpenLLaMA-3B student models on the corresponding supervised dataset with 10,000 steps. The GPT-2 teacher and student models use the fine-tuned checkpoints by Gu et al. [14]. For the implementation of compared baselines, we use the code by Ko et al. [21] and re-run the results. The optimization protocol for KD training largely follows the previous work [14, 21]. In particular, we search for the learning rates among a finite set for each experiment to obtain the best result. The batch size for each experiments is seleted to make full use of the 40GB RAM of an A40 GPU. To handle the adversarial training, we choose the number of adversarial steps $K = 5$ and the adversarial control factor $\alpha = 0.1$ based on the development experiments. We use a default off-/on-policy combination factor $\beta = 0.5$ for main experiments while exploring other values for analysis. The hyperparameters for training are listed in Table 3.

---

[2]`https://huggingface.co/openai-community/gpt2-xl`
[3]`https://huggingface.co/openai-community/gpt2`
[4]`https://huggingface.co/openlm-research/open_llama_7b`
[5]`https://huggingface.co/openlm-research/open_llama_7b`

Table 3: Hyperparameters for instruction-following experiments.

| Hyperparameter | GPT-2 | OpenLLaMA |
|---|---|---|
| Max. Step Size ($K$) | 10,000 | 10,000 |
| Inner Step Size ($N$) | 5 | 5 |
| Batch Size (per GPU) | {8, 16, 32} | {4, 8} |
| Dropout Rate | 0.1 | 0.1 |
| Controlling Factor ($\alpha$) | 0.1 | 0.1 |
| Discounting Factor ($\gamma$) | {0.90, 0.95, 0.99} | {0.90, 0.95, 0.99} |
| Combination Factor ($\beta$) | {0, 0.5, 0.9, 0.99, 1.0} | {0, 0.5, 0.9, 0.99, 1.0} |
| Learning Rate ($\eta$) | {$5e^{-5}$, $1e^{-4}$, $5e^{-4}$} | {$5e^{-6}$, $1e^{-5}$, $5e^{-5}$} |
| Warmup Steps | 1,000 | 500 |
| Weight Decay | $1e^{-2}$ | $1e^{-2}$ |
| Max Seq. Length | 512 | 512 |
| Sampling (top-p) | 1.0 | 1.0 |
| Sampling (temperature) | 1.0 | 1.0 |
| Evaluation | Greedy Sampling | Greedy Sampling |
| #GPUs | 2 | 4 |

## B.2 Task-Specific Experiments

**Base models.** For the text summarization and commonsense reasoning experiments, we use T5-XL[6] with 2.8B parameters to construct the teacher policy and construct the student policy with T5-Large[7] (770M parameters), T5-Base[8] (220M parameters) and T5-Small[9] (60M parameters). For the machine translation experiments, we use mT5-XL [43] to construct the teacher policy and mT5-Large/-Base/-Small to construct the student policy.

**Training details.** We initialize the corresponding teacher and student models using 10,000-step-fine-tuning checkpoints on the SAMSum dataset, 80,000-step-fine-tuning checkpoints on the IWSLT'17 (en-de) dataset and 3,000-step-fine-tuning checkpoints on the StrategyQA dataset. We largely follow Ko et al. [21] to set the hyperparameters for training. In particular, we search for the learning rate from a preset range to obtain the best result for each baseline and our method. The batch size is selected to make full use of the RAM of GPUs. We use a relatively larger maximum number of training steps for IWSLT'17 (en-de) experiments to satisfy sufficient convergences for the machine translation task. We use beam search for the evaluation of the IWSLT'17 (en-de) dataset.

Table 4: Hyperparameters for three task-specific experiments.

| Hyperparameter | SAMSum | IWSLT'17 (en-de) | StrategyQA |
|---|---|---|---|
| Max. Step Size ($K$) | 10,000 | 80,000 | 3,000 |
| Inner Step Size ($N$) | 5 | 2 | 5 |
| Batch Size (per GPU) | {16, 32, 64} | {16, 32, 64} | {16, 32, 64} |
| Dropout Rate | 0.0 | 0.3 | 0.1 |
| Controlling Factor ($\alpha$) | 0.1 | 0.1 | 0.1 |
| Discounting Factor ($\gamma$) | {0.90, 0.95, 0.99} | {0.90, 0.95, 0.99} | {0.90, 0.95, 0.99} |
| Combination Factor ($\beta$) | {0, 0.5, 0.9, 0.99, 1.0} | {0, 0.5, 0.9, 0.99, 1.0} | {0, 0.5, 0.9, 0.99, 1.0} |
| Learning Rate ($\eta$) | {$5e^{-5}$, $1e^{-4}$, $5e^{-4}$} | {$1e^{-4}$, $5e^{-4}$, $1e^{-3}$} | {$1e^{-4}$, $5e^{-4}$, $1e^{-3}$} |
| Warmup Steps | 1,000 | 4,000 | 300 |
| Weight Decay | $1e^{-2}$ | $1e^{-4}$ | $1e^{-2}$ |
| Max. Seq. Length | 1,024 | 512 | 1,024 |
| Sampling (top-p) | 1.0 | 1.0 | 1.0 |
| Sampling (temperature) | 1.0 | 1.0 | 1.0 |
| Evaluation | Greedy Sampling | Beam Search | Greedy Sampling |
| #GPUs | 2 | 4 | 1 |

---

[6]`https://huggingface.co/google/t5-v1_1-xl`

[7]`https://huggingface.co/google/t5-v1_1-large`

[8]`https://huggingface.co/google/t5-v1_1-base`

[9]`https://huggingface.co/google/t5-v1_1-small`

Table 5: Comparison with state-of-the-art KD methods on the instruction-following dataset using fine-tuned GPT-2 XL (1.5B) as the teacher model and fine-tuned GPT-2 (0.1B) as the student model. We format **the best**, the second best and worse than SFT results.

| Method | DollyEval | | SelfInst | | VicunaEval | | S-NI | UnNI |
|---|---|---|---|---|---|---|---|---|
| | GPT-4 | R-L | GPT-4 | R-L | GPT-4 | R-L | R-L | R-L |
| *GPT-2 XL (teacher)* | $45.5_{\pm 0.7}$ | $28.2_{\pm 0.8}$ | $34.7_{\pm 1.6}$ | $14.3_{\pm 0.2}$ | $32.7_{\pm 1.6}$ | $16.2_{\pm 0.3}$ | $27.6_{\pm 0.3}$ | $32.2_{\pm 0.3}$ |
| SFT *(student)* | $29.8_{\pm 1.2}$ | $23.4_{\pm 0.2}$ | $20.2_{\pm 0.7}$ | $10.3_{\pm 0.5}$ | $17.8_{\pm 0.9}$ | $14.6_{\pm 0.4}$ | $16.1_{\pm 0.3}$ | $18.2_{\pm 0.6}$ |
| KD [16] | $29.5_{\pm 0.8}$ | $23.8_{\pm 0.3}$ | $18.0_{\pm 1.0}$ | $12.3_{\pm 0.2}$ | $17.2_{\pm 0.7}$ | $15.2_{\pm 0.4}$ | $20.8_{\pm 0.5}$ | $22.5_{\pm 0.3}$ |
| SeqKD [20] | $29.8_{\pm 0.5}$ | $24.2_{\pm 0.2}$ | $18.2_{\pm 0.8}$ | $11.6_{\pm 0.4}$ | $18.2_{\pm 0.7}$ | $15.5_{\pm 0.3}$ | $15.5_{\pm 0.6}$ | $20.1_{\pm 0.1}$ |
| ImitKD [24] | $26.4_{\pm 0.6}$ | $22.7_{\pm 0.5}$ | $18.2_{\pm 0.5}$ | $11.5_{\pm 0.4}$ | $18.6_{\pm 0.4}$ | $14.5_{\pm 0.3}$ | $18.2_{\pm 0.3}$ | $21.8_{\pm 0.4}$ |
| MiniLLM [14] | $30.2_{\pm 1.2}$ | $24.3_{\pm 0.3}$ | $20.5_{\pm 0.3}$ | $13.2_{\pm 0.3}$ | $20.5_{\pm 0.7}$ | $18.5_{\pm 0.3}$ | $22.7_{\pm 0.3}$ | $23.5_{\pm 0.2}$ |
| GKD [2] | $29.2_{\pm 0.6}$ | $23.6_{\pm 0.2}$ | $20.7_{\pm 0.5}$ | $12.7_{\pm 0.2}$ | $20.2_{\pm 0.6}$ | $17.7_{\pm 0.2}$ | $25.1_{\pm 0.3}$ | $25.9_{\pm 0.1}$ |
| DistiLLM [21] | $31.2_{\pm 0.4}$ | $25.2_{\pm 0.4}$ | $21.7_{\pm 0.5}$ | $12.5_{\pm 0.3}$ | $22.5_{\pm 1.2}$ | $19.2_{\pm 0.5}$ | $27.7_{\pm 0.2}$ | $27.6_{\pm 0.4}$ |
| **Ours** | **$31.7_{\pm 0.5}$** | **$26.1_{\pm 0.3}$** | **$22.7_{\pm 0.5}$** | **$14.2_{\pm 0.3}$** | **$23.6_{\pm 0.8}$** | **$20.5_{\pm 0.2}$** | **$28.6_{\pm 0.2}$** | **$29.9_{\pm 0.5}$** |

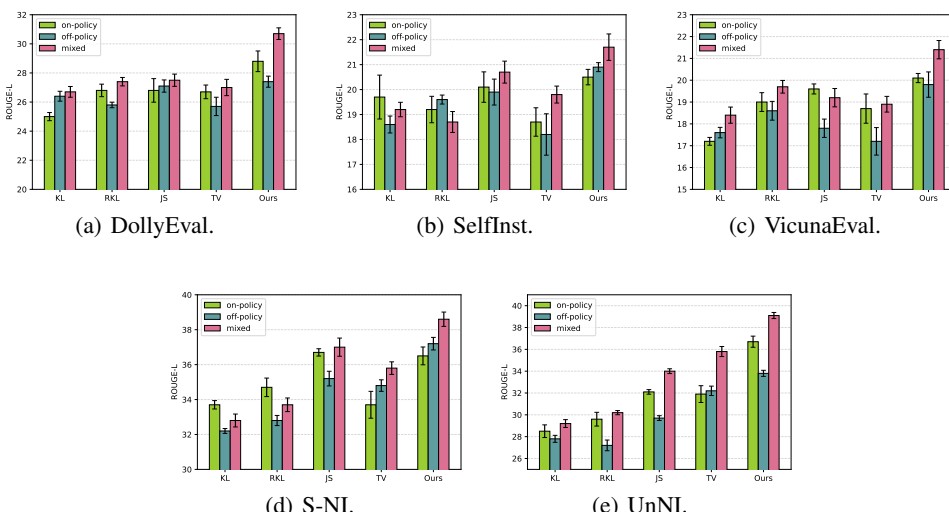

Figure 4: Performance of difference step-wise distribution distances on five instruction-following datasets using OpenLLaMA-7B → OpenLLaMA-3B.

## C  Additional Results

### C.1  Results Based on GPT-2

In addition to the experimental results based on OpenLLaMA for instruction-following tasks, we also conduct experiments based on GPT-2. Results are illustrated in Table 5. Compared with current state-of-the-art KD approaches, our method achieves the best results on five datasets with both GPT-4 feedback and ROUGE-L evaluations.

### C.2  Comparisons on Step-Wise Distribution Distance

Figure 4 and Figure 5 illustrate performance comparison with well-studied step-wise distribution distance, including KL, RKL, JS divergences and TV distances. Results show that the optimization of proposed moment-matching objectives outperforms other step-wise distribution distances via either on-policy distillation or off-policy distillation. Besides, jointly using on-policy and off-policy moment-matching further improves the performances and achieves the best results on five instruction-following datasets with KD from the OpenLLaMA-7B to OpenLLaMA-3B model, and achieves the best results on three task-specific datasets with KD from the (m)T5-XL to (m)T5-Base model.

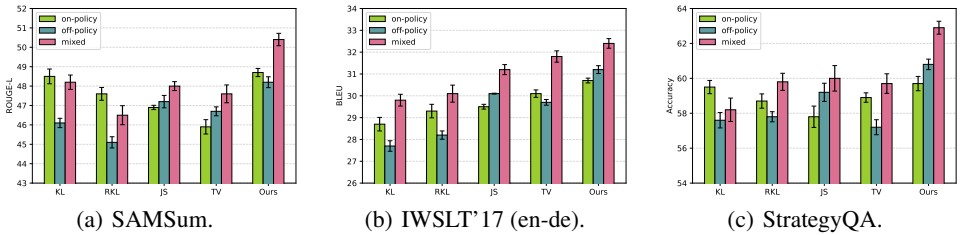

(a) SAMSum.  (b) IWSLT'17 (en-de).  (c) StrategyQA.

Figure 5: Performance of difference step-wise distribution distances on three task-specific datasets using (m)T5-XL → (m)T5-Base.

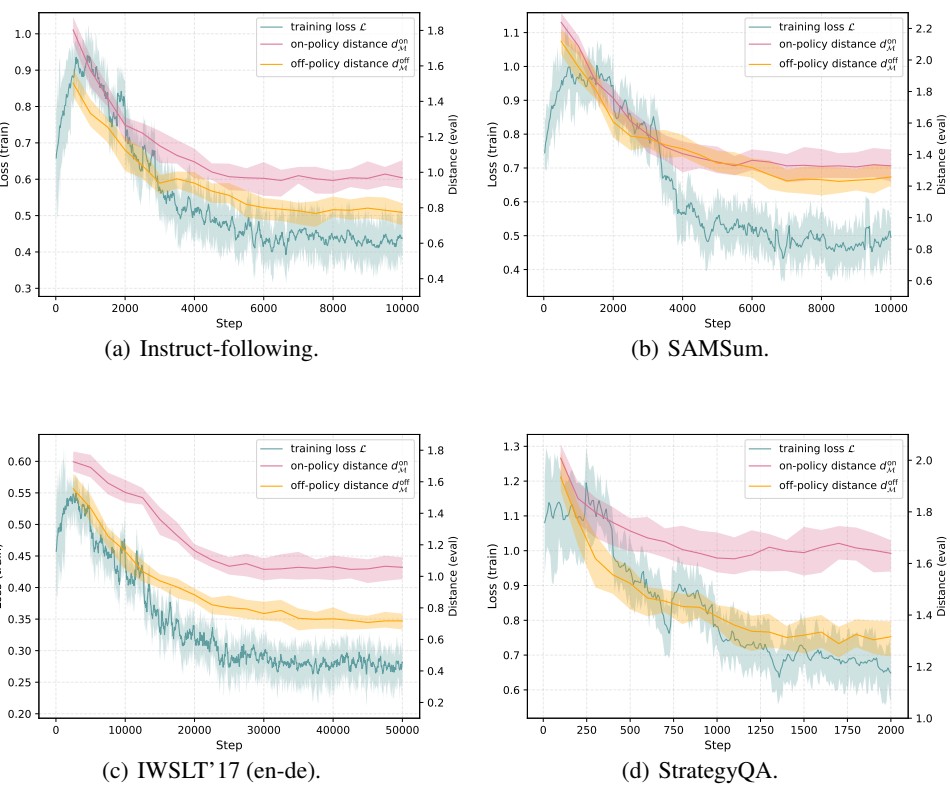

(a) Instruct-following.  (b) SAMSum.

(c) IWSLT'17 (en-de).  (d) StrategyQA.

Figure 6: Training loss and $d_{\mathrm{MM}}^{\mathrm{on}}$, $d_{\mathrm{MM}}^{\mathrm{off}}$ against training step on four datasets.

## C.3  Adversarial Training Procedure

Figure 6 illustrates the training loss and on-/off-policy moment-matching distances against the training steps on the instruction-following dataset and three task-specific datasets. We can observe that the training losses on four datasets have a similar trend, increasing at the beginning and then converging to a relatively lower level. The trend of loss function aligns with the characteristics of adversarial training with gradient descent ascent. In contrast, both the on-policy moment-matching distance $d_{\mathrm{MM}}^{\mathrm{on}}$ and the off-policy moment-matching distance $d_{\mathrm{MM}}^{\mathrm{off}}$ reduce as the number of training steps increases, which shows the effectiveness of our adversarial training approach for moment-matching.

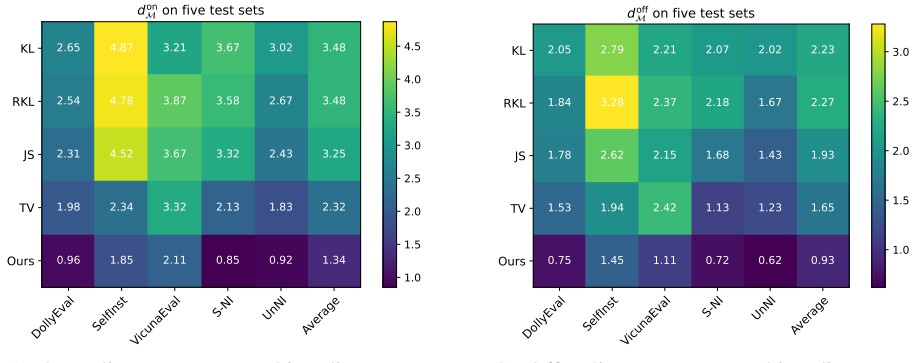

(a) On-policy moment-matching distance.  (b) Off-policy moment-matching distance.

Figure 7: Moment-matching via distribution-matching on the instruction-following dataset.

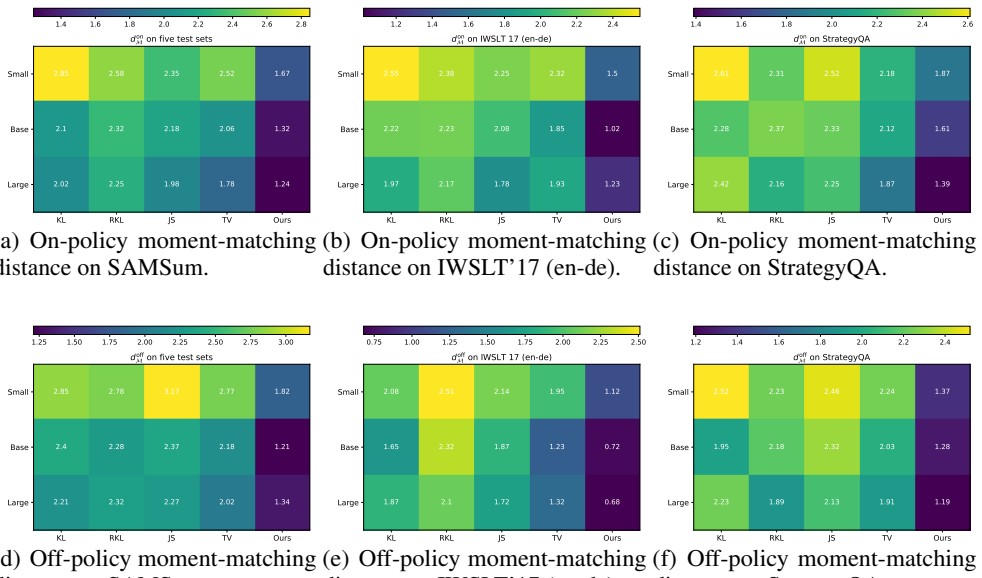

(a) On-policy moment-matching distance on SAMSum.  (b) On-policy moment-matching distance on IWSLT'17 (en-de).  (c) On-policy moment-matching distance on StrategyQA.

(d) Off-policy moment-matching distance on SAMSum.  (e) Off-policy moment-matching distance on IWSLT'17 (en-de).  (f) Off-policy moment-matching distance on StrategyQA.

Figure 8: Moment-matching via distribution-matching on three task-specific datasets.

## C.4   Moment-Matching via Distribution Matching

We investigate how the distribution-matching methods via KL, RKL, JS divergences or TV distance can optimize the moment-matching distance in Figure 7 and Figure 8. Results show that the proposed adversarial training algorithm is more effective in minimizing the moment-matching distance than the distribution-matching methods.

## C.5   Analysis on the Off-/On-Policy Combination Factor $\beta$

We study the impact of on-policy and off-policy objectives with the combination factor $\beta \in \{0.00, 0.25, 0.50, 0.75, 1.00\}$ in Eq. (9), which denotes a linear combination coefficient of the on-policy and off-policy objectives. We observe that if $\beta = 0.00$, only the on-policy objective contributes to policy learning. As it increases from 0 to 1, the influence of off-policy objective increases while that of the on-policy objective decreases. Finally, when $\beta = 1.00$, only the off-policy objective contributes to policy learning. We conduct experiments across four datasets. Specifically, we evaluate ROUGE-L for OpenLLaMA2-3B on the DollyEval dataset, ROUGE-L for T5-base on the SAMSum dataset, accuracy for T5-base on the IWSLT'17 dataset and accuracy for T5-base on

Table 6: Effects of the off-/on-policy combination factor $\beta$ on four datasets.

| $\beta$ | 0.00 | 0.25 | 0.50 | 0.75 | 1.00 |
|---|---|---|---|---|---|
| DollyEval | $28.8_{\pm 0.7}$ | $\mathbf{31.2}_{\pm 0.3}$ | $30.7_{\pm 0.4}$ | $29.8_{\pm 0.2}$ | $27.4_{\pm 0.4}$ |
| SAMSum | $48.2_{\pm 0.3}$ | $50.5_{\pm 0.2}$ | $50.4_{\pm 0.3}$ | $\mathbf{51.2}_{\pm 0.4}$ | $48.7_{\pm 0.2}$ |
| IWSLT'17 (en-de) | $30.7_{\pm 0.1}$ | $31.7_{\pm 0.6}$ | $32.4_{\pm 0.2}$ | $\mathbf{33.2}_{\pm 0.2}$ | $31.2_{\pm 0.2}$ |
| StrategyQA | $59.7_{\pm 0.4}$ | $61.4_{\pm 0.2}$ | $\mathbf{62.9}_{\pm 0.4}$ | $62.7_{\pm 0.4}$ | $60.8_{\pm 0.3}$ |

the StrategyQA dataset. Results in Table 6 show that a combination of on-policy and off-policy objectives outperforms using either on-policy or off-policy objectives only across four datasets.

