# OpenReview forum: "Adversarial Moment-Matching Distillation of Large Language Models"
_NeurIPS.cc/2024/Conference — NeurIPS 2024 poster_

### Official Review · Reviewer_ejno · 2024-07-03

**Soundness:** 3
**Presentation:** 3
**Contribution:** 2
**Rating:** 7
**Confidence:** 3

**Summary:**

To improve knowledge distillation for large language models, the authors first motivate an RL-based formulation that aims to minimize the imitation gap while matching on and off-policy moment bounds, and then introducing an adversarial training algorithm that achieves this by posing it as a two-player minimax game. They showcase the efficacy of the approach with instruction-following and task-specific experiments.

**Strengths:**

**S1.** The problem is an important one and, as far as I can tell, the method is novel and well motivated.

**S2.** The experiments are reasonable to establish the efficacy of the method.

**S3.** The paper is well written and clear.

**Weaknesses:**

**W1.** *The paper is missing a more thorough comparison of the training costs of each of the compared methods.* The authors briefly mention in Section 5 that their method induces a larger computational and memory cost compared to some of the other baselines. This discussion should be fleshed out more and potentially backed by experimental evidence.

Minor comments:
- It does not have any practical influence, but the definition of $\mathbf{y}$ in line 101 is not very elegant as $y_0$ will have a different dimensionality than $y_i$ for $i \in \\{1,\dots,T\\}$. It might be cleaner to define $\mathbf{y} = \mathbf{x} || (y_1,\dots,y_T)$, where $||$ is a concatenation operator.

**Questions:**

- Is the comparison between SFT and the other methods fair in terms of compute budget? It might be interesting to see if increasing the number of epochs still leads to a performance improvement, or if it has plateaued.
- Could there be any practical benefit to having a different $\alpha$ for on and off-policy optimization in Algorithm 1? The intuition for this comes from the results of Section 4.3 and particularly Figure 2.

**Limitations:**

The authors discuss the limitations appropriately in Section 5 of the paper.

---

> ### Author Rebuttal · Authors · 2024-08-07
>
> We sincerely appreciate your detailed and constructive comments, as well as your support of our work. We hope our responses adequately address your concerns.
>
> **Q1. Is the comparison between SFT and the other methods fair in terms of compute budget? It might be interesting to see if increasing the number of epochs still leads to a performance improvement, or if it has plateaued.**
> > Thanks for your suggestion! In our experiments, we use sufficient numbers of training steps to ensure our method and the baselines converge. The comparisons on all the datasets are based on the converged results. To investigate if increasing the iteration steps can lead to a performance improvement, we extended the training epochs to 4, with a maximum of 20,000 iteration steps in the instruction-following experiments. We evaluated our method, SFT, and two strong baselines using checkpoints from different training steps. The results show that (i) with sufficient training steps, e.g., $\geq 8,000$, all methods converge to a plateau with only small fluctuations, and (ii) under the premise of convergence and using the same amount of computation (e.g., training steps = 8,000, 10,000, 15,000, 20,000), our model still outperforms the other baselines.
> |Steps|	2,000|	4,000|	6,000|	8,000|	10,000|	15,000|	20,000|
> |----|----|----|----|----|----|----|----|
> |SFT|	14.7$_{\pm 0.5}$|	24.8$_{\pm 0.3}$|	26.8$_{\pm 0.3}$|	25.4$_{\pm 0.3}$|	26.7$_{\pm 0.5}$|	27.1$_{\pm 0.2}$|	26.6$_{\pm 0.2}$|
> GKD|	12.1$_{\pm 0.4}$|	20.7$_{\pm 0.4}$|	26.4$_{\pm 0.3}$|	27.0$_{\pm 0.4}$|	26.8$_{\pm 0.4}$|	27.3$_{\pm 0.3}$	|27.7$_{\pm 0.2}$|
> DistiLLM|	17.4$_{\pm 0.5}$|	25.9$_{\pm 0.3}$|	28.8$_{\pm 0.3}$|	29.7$_{\pm 0.3}$|	29.5$_{\pm 0.2}$|	28.9$_{\pm 0.2}$|	29.3$_{\pm 0.1}$|
> Ours|	16.9$_{\pm 0.6}$|	25.3$_{\pm 0.4}$|	29.5$_{\pm 0.5}$|	30.8$_{\pm 0.3}$|	30.7$_{\pm 0.2}$|	30.3$_{\pm 0.2}$|	30.6$_{\pm 0.2}$|
>
> **Q2. Could there be any practical benefit to having a different for on and off-policy optimization in Algorithm 1? The intuition for this comes from the results of Section 4.3 and particularly Figure 2.**
> > Thanks for your suggestion! We empirically investigated the learning rate controlling factors $\alpha$ for on-/off-policy $Q$-value function learning, respectively. We conducted instruction-following experiments with variants of off-policy $\alpha$ and on-policy $\alpha$ in ${0.05, 0.1, 0.15, 0.2} \times {0.05, 0.1, 0.15, 0.2}$ and illustrated the results on the DollyEval set in the table below. We observed that the best result (ROUGE-L = 31.5) was achieved with off-policy $\alpha = 0.15$ and on-policy $\alpha = 0.10$, while the $\underline{\text{second best}}$ result (ROUGE-L = 31.2) was achieved with off-policy $\alpha = 0.15$ and on-policy $\alpha = 0.05$, outperforming the results (ROUGE-L = 30.7) under equal $\alpha$ values for on- and off-policy optimization. This demonstrates the effects of using different $\alpha$ for on- and off-policy optimization.
> |On-Policy $\alpha$ ($\downarrow$) Off-policy $\alpha$ ($\rightarrow$) |	0.05|	0.10|	0.15|	0.20|
> |----|----|----|----|----|
> |**0.05**|	$30.5_{\pm 0.2}$|	$30.2 _{\pm 0.4}$|	$\underline{31.2}_{\pm 0.1}$|	$29.2_{\pm 0.2}$|
> |**0.10**|	$29.2_{\pm 0.5}$|	$30.7_{\pm 0.4}$|	${\bf 31.5}_{\pm 0.4}$|	$30.2_{\pm 0.5}$|
> |**0.15**|	$28.6_{\pm 0.4}$|	$29.6_{\pm 0.6}$|	$29.8_{\pm 0.4}$|	$28.8_{\pm 0.5}$|
> |**0.20**|	$27.9_{\pm 0.4}$|	$28.5_{\pm 0.4}$|	$28.2_{\pm 0.6}$|	$29.5_{\pm 0.6}$|
>
> In addition to addressing your insightful questions, we would also like to clarify the weakness as follows.
>
> **W1. The paper is missing a more thorough comparison of the training costs of each of the compared methods.**
> > Please refer to the [global response (Q2)](https://openreview.net/forum?id=0VeSCjRDBy&noteId=HgJrUiZdkn), where we analyze and compare the training costs, including both memory and computational complexities, of our method and the baselines. The analysis shows that our method does not significantly increase the memory or computational complexity in one iteration of policy updating.
>
> We also appreciate your comments on the minor weakness. We will incorporate your suggestions into our paper revision.

---

> > ### Comment · Reviewer_ejno · 2024-08-12
> >
> > I thank the authors for the additional results and appreciate their effort in the rebuttal. I believe these only serve to improve the strength of the paper (particularly the ones on the training costs), and will maintain my acceptance score.

---

### Official Review · Reviewer_zEWr · 2024-07-07

**Soundness:** 3
**Presentation:** 3
**Contribution:** 3
**Rating:** 6
**Confidence:** 3

**Summary:**

This paper applies a reinforcement learning (RL) framework to the problem of auto-regressive text generation, framing knowledge distillation as a task of minimizing the imitation gap between teacher and student policies. The authors provide a theoretical analysis demonstrating that the proposed momentum-matching method offers a tighter bound on this imitation gap compared to traditional distribution-matching approaches, potentially leading to improved optimization. To efficiently optimize the momentum-matching target, the paper introduces an adversarial training procedure that alternates updates between the student policy parameters and the Q-value functions, which are used to assess the imitation gaps. Experimental results showcased within the paper indicate that the proposed momentum-matching method outperforms existing distribution-matching baselines in terms of effectiveness.

**Strengths:**

1. The paper is well-structured and presents a clear, enjoyable narrative, facilitating ease of understanding for the reader.

2. Utilizing an RL framework, the paper theoretically demonstrates that the momentum-matching target provides a tighter bound for minimizing the imitation gap compared to conventional distribution-matching targets. Furthermore, an adversarial training procedure is proposed to effectively optimize the momentum-matching target, aiming to approximate a Nash equilibrium between the parameters of the student policy and the Q-value functions.

3. Comprehensive experiments empirically demonstrate that the proposed method outperforms existing distribution-matching methods in performance. Additionally, the presented analysis of training loss curves illustrates the stability of the proposed adversarial training procedure.

**Weaknesses:**

1. Unlike distribution-matching methods, the RL-based momentum-matching adversarial framework requires significantly more computational resources and runtime due to the necessity of calculating policy gradients and updating the parameters of auxiliary networks involved in Q-value functions. While the authors acknowledge this limitation in the conclusion section, the paper lacks quantitative analysis concerning the computation of policy gradients. It does not detail the resource consumption of the overall procedure. This omission limits the reader’s ability to assess the practical applicability of the method.

2. The presented experiments primarily examines knowledge distillation performance within similar or identical architectural frameworks. However, it does not demonstrate the method's generalizability across models with different architectures, thus leaving the robustness of the approach across diverse settings untested.

**Questions:**

1. Could you provide a detailed comparison of resource consumption and memory costs relative to the baseline methods? Additionally, can you discuss the impact of the number of samples used to estimate the policy gradient on the performance of your method?

2. It seems that the performance of the auxiliary models used to compute Q-value functions significantly impacts the effectiveness of your method. However, the paper does not provide details on the training of these models. I am curious to know whether these models are trained from scratch during the adversarial procedure or if they are pre-trained on certain datasets before inclusion. Is it possible to directly fine-tune teacher models to compute Q-value functions?

**Limitations:**

The paper briefly discusses the additional computational consumption and GPU memory costs associated with the proposed methods. However, it lacks a more thorough and detailed discussion of other potential limitations that may affect the applicability or scalability of the approach.

---

> ### Author Rebuttal · Authors · 2024-08-07
>
> We sincerely appreciate your detailed and constructive comments, as well as your support of our work. We hope our responses adequately address your concerns.
>
> **Q1. a. Could you provide a detailed comparison of resource consumption and memory costs relative to the baseline methods? b. Additionally, can you discuss the impact of the number of samples used to estimate the policy gradient on the performance of your method?**
> > **a. (Computational and Memory costs)** Thanks for your suggestion! Please refer to the [global response (Q2)](https://openreview.net/forum?id=0VeSCjRDBy&noteId=HgJrUiZdkn), where we analyze and compare the memory and computational complexities of our method with baseline methods. The analysis shows that our approach does not significantly increase memory or computational complexity in a single iteration of policy updating.
>
> > **b. (Impact of the Number of Samples)** Thanks for your suggestion! To analyze the impact of sample size on the performance of policy learning, we conducted experiments on an instruction-following dataset with varying numbers of samples. The ROUGE-L results, based on OpenLLaMA2-3B as the student model, are summarized below. They show that as the number of training samples increases, performance on the five test sets improves rapidly at first but then stabilizes once a certain sample size is reached. This demonstrates the importance of sample size in optimizing the policy gradient.
>
> |number of samples| 1,000| 2,000| 4,000| 6,000| 8,000| 10,000| 15,000|
> |----|----|----|----|----|----|----|----|
> |DollyEval|8.7$_{\pm 0.7}$ | 16.2$_{\pm 0.8}$ | 21.5$_{\pm 0.5}$ | 26.8$_{\pm 0.4}$ | 28.3$_{\pm 0.3}$ | 30.0$_{\pm 0.4}$| 30.7$_{\pm 0.4}$ |
> |SelfInst| 6.7$_{\pm 1.1}$ | 10.2$_{\pm 0.8}$ | 12.7$_{\pm 0.7}$ | 15.5$_{\pm 0.8}$ | 18.2$_{\pm 0.6}$ | 21.8$_{\pm 0.5}$ | 21.7$_{\pm 0.5}$ |
> |VicunaEval| 2.8$_{\pm 0.9}$  | 7.9$_{\pm 0.7}$  | 12.0$_{\pm 0.8}$ | 14.6$_{\pm 0.7}$ | 19.7$_{\pm 0.5}$ | 20.8$_{\pm 0.3}$ | 21.4$_{\pm 0.4}$ |
> |S-NI| 10.3$_{\pm 0.7}$ | 19.7$_{\pm 0.8}$ | 23.6$_{\pm 0.5}$ | 30.2$_{\pm 0.6}$ | 34.8$_{\pm 0.5}$ | 38.2$_{\pm 0.3}$ | 38.7$_{\pm 0.4}$ |
> |UnNI| 15.2$_{\pm 1.2}$ | 16.4$_{\pm 1.3}$ |22.7$_{\pm 0.6}$ | 30.6$_{\pm 0.8}$ | 37.2$_{\pm 0.5}$ | 38.3$_{\pm 0.6}$ | 39.1$_{\pm 0.3}$ |
>
> **Q2. a. Whether the $Q$-value functions are trained from scratch during the adversarial procedure or if they are pre-trained on certain datasets before inclusion. b. Is it possible to directly fine-tune teacher models to compute Q-value functions?**
> > **a. ($Q$-value Function Formulation)** Please refer to the [global response (Q1)](https://openreview.net/forum?id=0VeSCjRDBy&noteId=HgJrUiZdkn) for detailed formulations of the $Q$-value functions. These functions are constructed by adding linear layers to the student network and the parameters of these layers are trained from scratch during the adversarial procedure for Q-value function estimation.
>
> > **b.** Thanks for your suggestion! We agree that directly fine-tuning teacher models for $Q$-value function estimation is a novel idea. Our current approach uses an adversarial training strategy to estimate the $Q$-value function by solving a minimax optimization problem. Considering memory and computational complexity (as discussed in [global response Q2](https://openreview.net/forum?id=0VeSCjRDBy&noteId=HgJrUiZdkn)), we opted to train a linear layer on the student network for $Q$-value function estimation (as discussed in part **a**). However, fine-tuning the teacher model for $Q$-value function estimation, and subsequently using this $Q$-value for imitation learning, is an interesting direction that we may explore in future work.
>
> In addition to addressing your insightful questions, we would also like to clarify the remaining weaknesses of our work. W1 has been addressed in Q1, so we will focus on W2 below.
>
> **W2. The presented experiments primarily examines knowledge distillation performance within similar or identical architectural frameworks. However, it does not demonstrate the method's generalizability across models with different architectures, thus leaving the robustness of the approach across diverse settings untested.**
> > Thank you for your insightful suggestions! In our experiments, we have evaluated decoder-based LM architectures, such as OpenLLaMA2 and GPT-2, as well as encoder-decoder-based architectures like T5 and mT5. Due to limited computational resources, we were unable to evaluate the proposed approach on a wider range of architectures, such as OPT, LSTM, and GLM. However, we plan to explore these in future work.
>
> We also appreciate your comments on the weaknesses, and we will incorporate your suggestions into our paper revision.

---

### Official Review · Reviewer_8hQS · 2024-07-10

**Soundness:** 3
**Presentation:** 3
**Contribution:** 3
**Rating:** 7
**Confidence:** 4

**Summary:**

The paper introduces a novel approach to knowledge distillation for Large Language Models (LLMs) using an adversarial training method that incorporates both on and off-policy distillation. The method jointly learns a critic that estimates Q-values while updating both the Q-function and the student model to more closely match the teacher model. The authors employ a policy gradient method to update the student model.

**Strengths:**

+ As far as I am aware, a novel approach to knowledge distillation for LLMs -- although I am not an expert.
+ A well-presented method, tying together some previous ideas on IRL into the distillation application.
+ Demonstrates a boost to accuracy.

**Weaknesses:**

+ Some crucial details are unclear, particularly regarding the parameterization of the Q-value function (see questions section)
+ Not very much discussion of the computational complexity or additional overhead in memory of having multiple models and requiring rollouts from the teacher and student model while training
+ Lack of ablation studies -- the method is evaluated as a single monolithic method, when there are many variants that could be applied, such as a weighted combination of the two upper bounds. In particular, I'd like to see how the method using only the on-policy upper bound and the method using only the off-policy upper bound would compare against the method using the linear combination of the on and off policy upper bounds.

**Questions:**

+ How exactly is the Q-value function parameterized? Is it an extra head on the model, or a new model entirely?
+ Regarding the use of policy gradients for training the student function:
   a. Did you use a baseline to reduce variance?
   b. What is the variance of these policy gradients? In applications such as RL, policy gradients typically have quite high variance compared to other methods.
   c. Have you considered lightweight baseline methods, such as those presented in [1]?
+ Can you provide an ablation of the different elements of the approach, such as investigating the relative importance of the two upper bounds?
+ Can you provide an analysis -- even if it is brief -- on the computational cost and memory usage of using the additional Q-value critics while training?



[1] Ahmadian, Arash, et al. "Back to basics: Revisiting reinforce style optimization for learning from human feedback in llms." arXiv preprint arXiv:2402.14740 (2024).

**Limitations:**

Yes

---

> ### Author Rebuttal · Authors · 2024-08-07
>
> Thank you for taking the time to read our paper and for providing constructive comments. We hope our detailed responses will adequately address your concerns.
>
> **Q1. How exactly is the Q-value function parameterized? Is it an extra head on the model, or a new model entirely?**
> > Please refer to the [global response (Q1)](https://openreview.net/forum?id=0VeSCjRDBy&noteId=HgJrUiZdkn), where we discuss the parameterization of the $Q$-value function, which is based on an additional trainable linear head on the student model.
>
> **Q2. Regarding the use of policy gradients for training the student function: a. Did you use a baseline to reduce variance? b. What is the variance of these policy gradients? In applications such as RL, policy gradients typically have quite high variance compared to other methods. c. Have you considered lightweight baseline methods, such as those presented in [1]?**
> > **ab. (Variance of Policy Gradient)** Since the student model was pre-trained with SFT, the variance of the policy gradient is sufficiently low to ensure stable training. Therefore, in consideration of computational efficiency, we did not employ a baseline model in our original experiments.
>
> > **c. (Baseline Model)** Thanks for your suggestion! Following the approach of Ahmadian et al. [1], we implemented two types of baseline models to reduce the variance of policy gradients. Specifically, the sampled baseline uses an average of $k=4$ trajectories draws to estimate the baseline, represented as $\hat{b} = \frac{1}{k} \sum_{i=1}^k U^{\rm on}(\tau_i, f)$, $\tau_1, \ldots, \tau_k \sim \pi_\theta(\cdot | x)$. The learned baseline involves optimizing a value network (a linear head) $b_\psi$ with respect to the MLE loss $\sum_s \vert U^{\rm on}(\tau_s, f) - b_{\psi}(\tau_s) \vert^2 $. Reuslts show that using a baseline can slightly improve performance, although it increases the computational complexity.
> |Dataset|	w/o baseline|	sampled baseline|	learned baseline|
> |----|----|----|----|
> |DollyEval (OpenLLaMA2-3B,ROUGE-L)|	27.4$_{\pm 0.4}$|	**27.8**$_{\pm 0.3}$|	26.8$_{\pm 0.2}$|
> SAMSum (T5-base, ROUGE-L )|	48.2$_{\pm 0.3}$|	**48.7**$_{\pm 0.2}$|	48.5$_{\pm 0.4}$|
> IWSLT’17 (T5-base, BLEU)|	31.2$_{\pm 0.2}$|	30.7$_{\pm 0.2}$|	**31.8**$_{\pm 0.1}$|
> StrategyQA (T5-base, Acc)|	60.8$_{\pm 0.3}$|	**61.2**$_{\pm 0.3}$|	58.5$_{\pm 0.3}$|
>
> >[1] Ahmadian, Arash, et al. "Back to basics: Revisiting reinforce style optimization for learning from human feedback in llms." arXiv preprint arXiv:2402.14740 (2024).
>
> **Q3. Can you provide an ablation of the different elements of the approach, such as investigating the relative importance of the two upper bounds?**
> > Thanks for your suggestion! Please refer to the [global response (Q3)](https://openreview.net/forum?id=0VeSCjRDBy&noteId=HgJrUiZdkn), where we present an ablation study on the effectiveness of the two upper bounds, using a linear combination of on-policy and off-policy objectives. The results demonstrate stronger performance compared to using either the on-policy or off-policy objective alone.
>
> **Q4. Can you provide an analysis -- even if it is brief -- on the computational cost and memory usage of using the additional Q-value critics while training?**
> > Thanks for your suggestion! Please refer to the [global response (Q2)](https://openreview.net/forum?id=0VeSCjRDBy&noteId=HgJrUiZdkn), where we analyze the computational and memory costs associated with using additional $Q$-value critics during training. Comparisons with baseline methods show that our method does not significantly increase memory or computational complexity for policy learning with the additional Q-value critics.
>
> **Thank you once again for your effort in providing detailed and constructive comments. If you have further questions, please feel free to discuss them with us. We will consider your comments and suggestions in our paper revision.**

---

> > ### Comment · Reviewer_8hQS · 2024-08-07
> > **Response to Author Rebuttal**
> >
> > Thank you for your thorough rebuttal.
> > Thanks for addressing all of my queries -- I know that probably took a lot of work, given the short timeframe.
> > Given that all the additional results raise no additional problems, and the remarks on the computational complexity illustrate that the overhead is relatively small, I will raise my score to 7.

---

### Official Review · Reviewer_M71H · 2024-07-12

**Soundness:** 3
**Presentation:** 3
**Contribution:** 3
**Rating:** 5
**Confidence:** 2

**Summary:**

This paper proposed an adversarial moment-matching approach for knowledge distillation of LLM. The idea is to reformulate the knowledge distillation from an imitation learning perspective and derive both on-policy and off-policy bounds for the imitation gap between the teacher and student models. The authors proposed an adversarial training algorithm to estimate and minimize the on-policy and off-policy moment-matching distances. The moment-matching distance is evaluated by the value function and the student is updated using policy gradients to minimize this distance. Experiments on instruction-following and task-specific datasets show that the proposed approach outperforms other knowledge distillation methods.

**Strengths:**

It is novel to reformulate the knowledge distillation as a moment matching problem, where the matching distance is evaluated by the Q-value function.

The authors derive the imitation gap bound for both on-policy and off-policy setup, and optimize the gap to achieve the knowledge distillation.

The proposed method demonstrated good performance on both instruction-following and task-specific datasets. The seven baselines are either distribution matching based or supervised finetuned.

I found the connection between the proposed moment-matching approach and distribution distances matching interesting. Specifically, the authors show that minimizing the total variation distance can achieve sub-optimal results for the moment-matching bounds

**Weaknesses:**

1. I would recommend having ablation studies and analysis of the impact of on-policy and off-policy objective. And analysis how each of them effect the overall performance.

2. Solving Eq.(9) is involving optimizing the minmax problem. (a) First, the optimization requires additional computational steps for the inner-loop gradient update. How expensive is the computation? such as the time/memory cost (b) Is the optimization robust with respect of hyperparameter changes like $K$ and $\alpha$.

3. The method requires an auxiliary network for Q-value estimation. It make the training system even more delicated. What network is used for Q-value estimation? any analysis here?

**Questions:**

Overall, I think the idea is novel. However, as authors pointed out in the limitation part, the required time/memory cost/training efforts can assumed to be high.  I would recommend have a comparison here with other distribution matching methods or knowledge distillation methods.

See the weakness section for my questions.

**Limitations:**

See above question and weakness sections.

---

> ### Author Rebuttal · Authors · 2024-08-07
>
> Thank you for taking the time to read our paper and for providing constructive comments. We hope our detailed responses will adequately address your concerns.
>
> **W1. Ablation studies and analysis of the impact of on-policy and off-policy objective. And analysis how each of them effect the overall performance.**
> > Thanks for your valuable suggestion! Please refer to the [global response (Q3)](https://openreview.net/forum?id=0VeSCjRDBy&noteId=HgJrUiZdkn), where we provide an analysis of a linear combination of on-policy and off-policy objectives, which demonstrates stronger performance compared to using only on-policy or off-policy objectives individually. Additionally, we analyze the impact of on-policy and off-policy objectives on other distribution-matching baselines, as depicted in Figure 2 of the paper.
>
> **W2. Solving Eq.(9) is involving optimizing the minmax problem. (a) First, the optimization requires additional computational steps for the inner-loop gradient update. How expensive is the computation? such as the time/memory cost (b) Is the optimization robust with respect of hyperparameter changes like $K$ and $\alpha$.**
>
> >**a. (Time/Memory Cost)** Please refer to the [global response (Q2)](https://openreview.net/forum?id=0VeSCjRDBy&noteId=HgJrUiZdkn), where we analyze and compare our method with baseline methods in terms of both memory and computational complexities. Our analysis indicates that our method does not significantly increase memory or computational complexity in one iteration of policy updating.
>
> >**b. (Robustness w.r.t. $K$ and $\alpha$)** In our adversarial training algorithm, the number of inner-loop steps $K$ and the learning rate controlling factor $\alpha$ are hyperparameters that control the stochastic gradient ascent learning of the on-/off-policy $Q$-value functions. To demonstrate the robustness of $K$ and $\alpha$ in policy optimization, we experimented with different values: $K \in {1, 3, 5, 10, 15}$ and $\alpha \in {0.001, 0.01, 0.1, 0.2, 0.5, 1.0}$ for the instruction-following experiments. The results on DollyEval are shown below. The results indicate that as the number of inner-loop steps $K$ increases, the estimation of on-/off-policy $Q$-value functions becomes more effective, contributing to knowledge distillation. After the training objective converges, increasing $K$ further, for instance, $K \geq 10$, does not significantly improve performance. A small controlling factor $\alpha$, for instance, $\alpha=0.001$ leads to a small learning rate for the $Q$-value function, i.e., $\alpha \eta$, making the $Q$-value in the minimax optimization too weak to serve as an effective critic for policy imitation learning. As $\alpha$ increases to an appropriate level that balances the the $Q$-value critic and the policy, distillation performance improves. However, a large $\alpha$, for example, $\alpha \geq 0.5$, can cause the $Q$-value critic to become too dominant, hindering effective policy learning during the adversarial training process, and leading to poor performance.
> >|$K$|	1|	3|	5|	10|	15|
> |----|----|----|----|----|----|
> |ROUGE-L|	28.5$_{\pm 0.4}$|	29.8$_{\pm 0.4}$|	30.7$_{\pm 0.4}$|	30.9$_{\pm 0.6}$|	30.2$_{\pm 0.4}$|
>
> >|$\alpha$|	0.001|	0.01|	0.1|	0.2|	0.5|	1.0|
> |----|----|----|----|----|----|----|
> |ROUGE-L|	27.7$_{\pm 0.5}$|	29.6$_{\pm 0.5}$|	30.7$_{\pm 0.4}$|	29.5$_{\pm 0.6}$|	28.4$_{\pm 0.4}$|	25.5$_{\pm 0.6}$|
>
> **W3. The method requires an auxiliary network for Q-value estimation. It make the training system even more delicated. What network is used for Q-value estimation? any analysis here?**
> > Please refer to the [global response (Q1)](https://openreview.net/forum?id=0VeSCjRDBy&noteId=HgJrUiZdkn), where we provide a detailed discussion of the network used for $Q$-value function estimation and analyze the associated memory and computational complexities.
>
> **Thanks again for your effort in giving detailed and constructive comments. If you have more questions, feel free to discuss them with us. We will take your comments and suggestions in paper revision.**

---

### Author Rebuttal · Authors · 2024-08-07

## Global Response to Common Questions
**Q1. The parameterized estimation of Q-value function.**
> The Q-value function $f_{\phi}(y_{<t}, y)$, where $y_{<t}$ denotes the state of the $0:t-1$ tokens and $y \in \mathcal{V}$ denotes an action of the $t$-th token, was estimated with a neural network, which can be represented as $f_{\phi}(y_{<t}, y) = (h_{t-1} + v_y)w_y^{\top}$. Here, $h_{t-1}$ denotes the feature representation of the state $y_{<t}$ derived from the $(t-1)$-th hidden step of an auto-regressive LM, $v_y$ represents the feature vector corresponding to the token $y$, while $w_y$ denotes the parameters of a linear layer. Thus, the $Q$-value function can be viewed as an additional linear layer on the model with the trainable parameters $\phi = ( v_y, w_y )_{y \in \mathcal{V}}$. Notably, the trainable parameters of the $Q$-value function were optimized starting from randomly initialized weights.

**Q2. Computational and memory costs of the minimax optimization with the Q-value estimation.**
>**a. (Memory Cost)** Compared to the baseline SFT and distribution-matching methods, the additional trainable parameters come from the $Q$-value function, which consists of $\mathcal{O}(H|\mathcal{V}|)$ parameters based on the response to **Q1**, where $H$ represents the hidden dimension and $|\mathcal{V}|$ represents the vocabulary size.

>**b. (Computational Cost)** For each iteration (one step of policy update) of our method, the computational complexity of our gradient-based algorithm (Alg. 1) consists of **(i)** the inner-loop gradient computation and updating: $\mathcal{O}(KT|\mathcal{V}|H)$, which is a multiple of $K$ steps in the inner-loop, the maximum number of sequence length $T$, the vocabulary size $|\mathcal{V}|$ and the hidden size $H$, and **(ii)** the policy gradient computation and updating: $\mathcal{O}(T|\mathcal{V}||\theta|)$, where $|\theta|$ represents the number of parameters in a LM. We observe that the computational complexity in other distribution-matching baselines, such as KL-, RKL-, JS-divergence and TV distance is equal to (ii) in our method. Thus, our method incurs additional computational complexity of (i) in compared with the baseline distribution-matching methods. Since the number of parameters in language model $|\theta|$ is much larger than the hidden dimension, e.g., $117M=|\theta|\gg KH= 3,840$ for GPT-2, $2.7B=|\theta|\gg KH= 16,000$ for OpenLLaMA2 and $220M=|\theta|\gg KH= 3,840$ for T5-base, **our method does not significantly increase the computational complexity in one iteration of policy gradient updating**.
||ours|baselines|
|----|----|----|
|memory complexity| $\mathcal{O}(H\vert\mathcal{V}\vert + \vert  \theta \vert)$ | $\mathcal{O}(\vert  \theta \vert)$ |
|computational complexity| $\mathcal{O}(KT\vert\mathcal{V}\vert H + T\vert\mathcal{V}\vert\vert\theta\vert)$ |  $\mathcal{O}(T\vert\mathcal{V}\vert\vert\theta\vert)$|

**Q3. Ablation study corresponding to the on- and off-policy learning.**
> We study the impact of on-policy and off-policy objectives within our method by defining the following linear combination policy gradient updating role: $ \theta \leftarrow \theta - 2\eta ( \gamma G_{\rm on}(\tau_{\rm on}, \theta) + (1-\gamma) G_{\rm off}(\tau_{\rm off}, \theta) ) $, where $\gamma \in [0,1]$ denotes the linear combination coefficient of the on-policy gradient $G_{\rm on}(\tau_{\rm on}, \theta)$ and the off-policy gradient $G_{\rm off}(\tau_{\rm off}, \theta)$. We observe that if $\gamma=0$, only the off-policy objective contributes to policy learning. As $\gamma$ increases from 0 to 1, the influence of on-policy objective increases while that of the off-policy objective decreases. Finally, when $\gamma=1$, only the on-policy objective contributes to policy learning. The following results shows that a combination of on-policy and off-policy objectives outperforms using either on-policy or off-policy objectives only across four datasets
|$\gamma$|0 (off-policy)|0.25|0.50|0.75|1.00 (on-policy)|
|----|----|----|----|----|----|
|DollyEval (OpenLLaMA2-3B, ROUGE-L)|27.4$_{\pm 0.4}$|	28.2$_{\pm 0.3}$|	**30.7**$_{\pm 0.4}$|	29.5$_{\pm 0.7}$|	28.8$_{\pm 0.7}$|
SAMSum (T5-base, ROUGE-L)|	48.2$_{\pm 0.3}$|	**50.8**$_{\pm 0.3}$|	50.4$_{\pm 0.3}$|	49.2$_{\pm 0.2}$|	48.7$_{\pm 0.2}$|
IWSLT’17 (T5-base, BLEU)|	31.2$_{\pm 0.2}$|	30.3$_{\pm 0.2}$|	**32.4**$_{\pm 0.2}$|	31.6$_{\pm 0.1}$|	30.7$_{\pm 0.1}$|
StrategyQA (T5-base, Acc)|	60.8$_{\pm 0.3}$|	**63.6**$_{\pm 0.5}$|	62.9$_{\pm 0.4}$|	60.5$_{\pm 0.5}$|	59.7$_{\pm 0.4}$|

---

### Decision · Program_Chairs · 2024-09-25

**Decision:**

Accept (poster)

**Comment:**

This paper presents a RL framework to the problem of auto-regressive text generation, framing knowledge distillation of LLMs as a task of minimizing the imitation gap between teacher and student policies. After author rebuttal, it received scores of 5677. All the reviewers are generally happy about the paper, agreeing that (1) the proposed method is novel, (2) the paper is well written, and (3) the experiments are reasonable to establish the efficacy of the method.

Reviewers had also shown some common concerns, such as clarity regarding the parameterization of the Q-value function, and lack of quantitative analysis concerning the computation of policy gradients. The authors have done a good job in addressing these concerns during rebuttal. Overall, the AC would like to recommend accept of the paper.